# AGENT-BASED GRAPH NEURAL NETWORKS

**Karolis Martinkus**[1], **Pál András Papp**[2], **Benedikt Schesch**[1], **Roger Wattenhofer**[1]
[1]ETH Zurich [2]Computing Systems Lab, Huawei Zurich Research Center

## ABSTRACT

We present a novel graph neural network we call AgentNet, which is designed specifically for graph-level tasks. AgentNet is inspired by sublinear algorithms, featuring a computational complexity that is independent of the graph size. The architecture of AgentNet differs fundamentally from the architectures of traditional graph neural networks. In AgentNet, some trained *neural agents* intelligently walk the graph, and then collectively decide on the output. We provide an extensive theoretical analysis of AgentNet: We show that the agents can learn to systematically explore their neighborhood and that AgentNet can distinguish some structures that are even indistinguishable by 2-WL. Moreover, AgentNet is able to separate any two graphs which are sufficiently different in terms of subgraphs. We confirm these theoretical results with synthetic experiments on hard-to-distinguish graphs and real-world graph classification tasks. In both cases, we compare favorably not only to standard GNNs but also to computationally more expensive GNN extensions.

## 1 INTRODUCTION

Graphs and networks are prominent tools to model various kinds of data in almost every branch of science. Due to this, graph classification problems also have a crucial role in a wide range of applications from biology to social science. In many of these applications, the success of algorithms is often attributed to recognizing the presence or absence of specific substructures, e.g. atomic groups in case of molecule and protein functions, or cliques in case of social networks [10; 77; 21; 23; 66; 5]. This suggests that some parts of the graph are "more important" than others, and hence it is an essential aspect of any successful classification algorithm to find and focus on these parts.

In recent years, Graph Neural Networks (GNNs) have been established as one of the most prominent tools for graph classification tasks. Traditionally, all successful GNNs are based on some variant of the message passing framework [3; 69]. In these GNNs, all nodes in the graph exchange messages with their neighbors for a fixed number of rounds, and then the outputs of all nodes are combined, usually by summing them [27; 52], to make the final graph-level decision.

It is natural to wonder if all this computation is actually necessary. Furthermore, since traditional GNNs are also known to have strong limitations in terms of expressiveness, recent works have developed a range of more expressive GNN variants; these usually come with an even higher computational complexity, while often still not being able to recognize some simple substructures. This complexity makes the use of these expressive GNNs problematic even for graphs with hundreds of nodes, and potentially impossible when we need to process graphs with thousands or even more nodes. However, graphs of this size are common in many applications, e.g. if we consider proteins [65; 72], large molecules [79] or social graphs [7; 5].

In light of all this, we propose to move away from traditional message-passing and approach graph-level tasks differently. We introduce AgentNet – a novel GNN architecture specifically focused on these tasks. AgentNet is based on a collection of trained neural agents, that intelligently *walk* the graph, and then collectively classify it (see Figure 1). These agents are able to retrieve information from the node they are occupying, its neighboring nodes, and other agents that occupy the same node. This information is used to update the agent's state and the state of the occupied node. Finally, the agent then chooses a neighboring node to transition to, based on its own state and the state of the neighboring nodes. As we will show later, even with a very naive policy, an agent can already recognize cliques and cycles, which is impossible with traditional GNNs.

---

Correspondence to `martinkus@ethz.ch`.

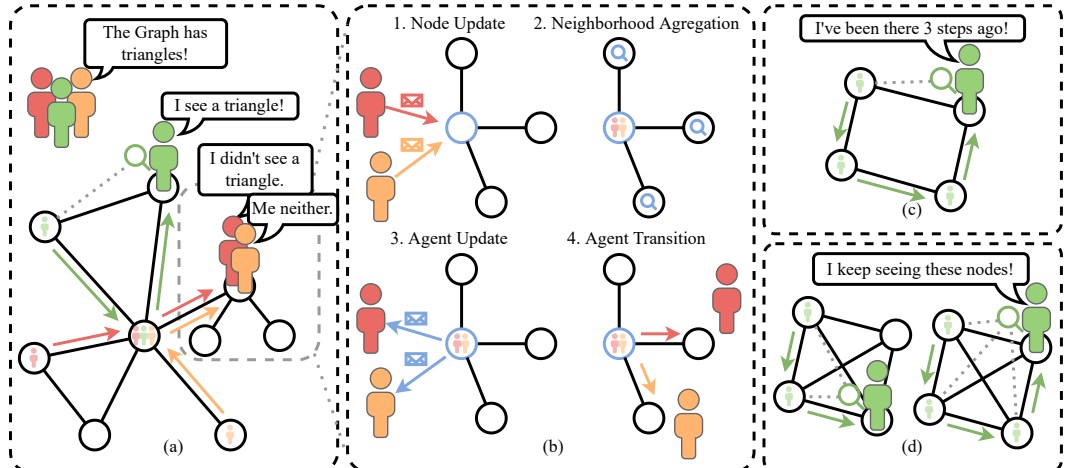

Figure 1: AgentNet architecture. We have many neural agents walking the graph (a). Each agent at every step records information on the node, investigates its neighborhood, and makes a probabilistic transition to another neighbor (b). If the agent has walked a cycle (c) or a clique (d) it can notice.

One of the main advantages of AgentNet is that its computational complexity only depends on the node degree, the number of agents, and the number of steps. This means that if a specific graph problem does not require the entire graph to be observed, then our model can often solve it using less than $n$ operations, where $n$ is the number of nodes. The study of such sublinear algorithms is a popular topic in graph mining [35; 26]; it is known that many relevant tasks can be solved in a sublinear manner. For example, our approach can recognize if one graph has more triangles than another, or estimate the frequency of certain substructures in the graph – in sublinear time!

AgentNet also has a strong advantage in settings where e.g. the relevant nodes for our task can be easily recognized based on their node features. In these cases, an agent can learn to walk only along these nodes of the graph, hence only collecting information that is relevant to the task at hand. The amount of collected information increases linearly with the number of steps. In contrast to this, a standard message-passing GNN always (indirectly) processes the entire multi-hop neighborhood around each node, and hence it is often difficult to identify the useful part of the information from this neighborhood due to oversmoothing or oversquashing effects [46; 2] caused by an exponential increase in aggregated information with the number of steps. One popular approach that can partially combat this has been attention [70] as it allows for soft gating of node interactions. While our approach also uses attention for agent transition sampling, the transitions are hard. More importantly, these agent transitions allow for the reduction of computational complexity and increase model expressiveness, both things the standard attention models do not provide.

## 2 RELATED WORK

### 2.1 GNN LIMITATIONS

**Expressiveness.** Xu et al. [75] and Morris et al. [53] established the equivalence of message passing GNNs and the first order Weisfeiler-Lehman (1-WL) test. This spurred research into more expressive GNN architectures. Sato et al. [64] and Abboud et al. [1] proposed to use random node features for unique node identification. As pointed out by Loukas [48], message passing GNNs with truly unique identifiers are universal. Unfortunately, such methods generalize poorly to new graphs [59]. Vignac et al. [71] propose propagating matrices of order equal to the graph size as messages instead of vectors to achieve a permutation equivariant unique identification scheme. Other possible expressiveness-improving node feature augmentations include distance encoding [45], spectral features [4; 25] or orbit counts [11]. However, such methods require domain knowledge to choose what structural information to encode. Pre-computing the required information can also be quite expensive [11]. An alternative to this is directly working with higher-order graph representations [53; 51], which can directly bring $k$-WL expressiveness at the cost of operating on $k$-th order graph representations. To improve this, methods that consider only a part of the higher-order interactions have been proposed [55; 56].

Alternatively, extensions to message passing have been proposed, which involve running a GNN on many, slightly different copies of the same graph to improve the expressiveness [59; 8; 80; 19; 62]. Papp & Wattenhofer [58] provide a theoretical expressiveness comparison of many of these GNN extensions. Geerts & Reutter [33] propose the use of tensor languages for theoretical analysis of GNN expressive power. Interestingly enough, all of these expressive models focus mainly on graph-level tasks. This suggests these are the main tasks to benefit from increased expressiveness.

**Scalability.** As traditional GNN architectures perform computations on the neighborhood of each node, their computational complexity depends on the number of nodes in the graph and the maximum degree. To enable GNN processing of large graphs Hamilton et al. [37] and Chen et al. [13] propose randomly sub-sampling neighbors every batch or every layer. To better preserve local neighborhoods Chiang et al. [16] and Zeng et al. [78] use graph clustering, to construct small subgraphs for each batch. Alternatively, Ding et al. [22] quantize node embeddings and use a small number of quantized vectors to approximate messages from out-of-batch nodes. In contrast to our approach, these methods focus on node-level tasks and might not be straightforward to extend to graph-level tasks, because they rely on considering only a predetermined subset of nodes from a graph in a given batch.

## 2.2 Sublinear algorithms

Sublinear algorithms aim to decide a graph property in much less than $n$ time, $n$ being the number of nodes [35]. It is possible to check if one graph has noticeably more substructures such as cliques, cycles, or even minors than another graph in constant time [31; 30; 6]. It is also possible to estimate counts of such substructures in sublinear time [26; 14; 7]. All of this can be achieved either by performing random walks [14; 7] or by investigating local neighborhoods of random nodes [31; 30; 6]. We will show that AgentNet can do both, local investigations and random walks.

## 2.3 Random walks in GNNs

Random walks have been previously used in graph machine learning to construct node and graph representations [61; 36; 67; 32; 57]. However, traditionally basic random walk strategies are used, such as choosing between dept-first or breath-first random neighborhood traversals [36]. It has also been shown by Geerts [32] that random-walk-based GNNs are at most as expressive as 2-WL test; we will show that our approach does not suffer from this limitation. Toenshoff et al. [67] have proposed to explicitly incorporate node identities and neighbor relations into the random walk trace to overcome this limitation. AgentNet is able to automatically learn to capture such information; moreover, our agents can learn to specifically focus on structures that are meaningful for the task at hand, as opposed to the purely random walks of [67]. As an alternative, Lee et al. [44] have proposed to learn a graph walking procedure using reinforcement learning. In contrast to our work, their approach is not fully differentiable; furthermore, the node features cannot be updated by the agents during the walk, and the neighborhood information is not considered in each step, which both are crucial for expressiveness.

## 3 AgentNet model

The core idea of our approach is that a graph-level prediction can be made by intelligently walking on the graph many times in parallel. To be able to learn such intelligent graph walking strategies through gradient descent we propose the AgentNet model (Figure 1). On a high level, each agent $a$ is assumed to be uniquely identifiable (have an ID) and is initially placed uniformly at random in the graph. Upon an agent visiting a node $v$ at time step $t$ (including the initial placement), the model performs four steps: node update, neighborhood aggregation, agent update, and agent transition.

First, the embedding of each node $v_i$ that has agents $a_j \in A(v_i)$ on it is updated with information from all of those agents using the node update function $f_v$. To ensure that the update is expressive enough, agent embeddings are processed before aggregation using a function $\phi_{a \to v}$:

$$v_i^t = f_v \left( v_i^{t-1}, \sum_{a_j^{t-1} \in A(v_i)} \phi_{a \to v}(a_j^{t-1}) \right) \quad \textbf{if } |A(v_i)| > 0 \textbf{ else } v_i^{t-1}.$$

Next, in the neighborhood aggregation step, current neighborhood information is incorporated into the node state using the embeddings of neighboring nodes $v_j \in N(v_i)$. Similarly, we apply a function

$\phi_{N(v) \to v}$ on the neighbor embeddings beforehand, to ensure the update is expressive:

$$v_i^t = f_n \left( v_i^t, \sum_{v_j^t \in N(v_i)} \phi_{N(v) \to v}(v_j^t) \right) \quad \textbf{if } |A(v_i)| > 0 \textbf{ else } v_i^t.$$

These two steps are separated so that during neighborhood aggregation a node would receive information about the current state of the neighborhood and not its state in the previous step. The resulting node embedding is then used to update agent embeddings so that the agents know the current state of the neighborhood (e.g. how many neighbors of $v_i$ have been visited) and which other agents are on the same node. With $V(a_i)$ being a mapping from agent $a_i$ to its current node:

$$a_i^t = f_a \left( a_i^{t-1}, v_{V(a_i)}^t \right).$$

Now the agent has collected all possible information at the current node it is ready to make a transition to another node. For simplicity, we denote the potential next positions of agent $a_i$ by $N^t(a_i) := N(v_{V(a_i)}^t) \cup v_{V(a_i)}^t$. First, probability logits $z_{a_i \to v_j}$ are estimated for each $v_j^t \in N^t(a_i)$. Then this distribution is sampled to select the next agent position $V(a_i)$:

$$z_{a_i \to v_j} = f_p \left( a_i^t, v_j^t \right) \quad \text{for} \quad v_j^t \in N^t(a_i),$$

$$V(a_i) \leftarrow \text{GumbelSoftmax} \left( \left\{ z_{a_i \to v_j} \quad \text{for} \quad v_j^t \in N^t(a_i) \right\} \right).$$

As we need a categorical sample for the next node position, we use the straight-through Gumbel softmax estimator [41; 50]. In the practical implementation, we use dot-product attention [68] to determine the logits $z_{a_i \to v_j}$ used for sampling the transition. To make the final graph-level prediction, pooling (e.g. sum pooling) is applied on the agent embeddings, followed by a readout function (an MLP). Edge features can also be trivially incorporated into this model. For details on this and the practical implementation of the model see Appendix C. There we also discuss other simple extensions to the model, such as providing node count to the agents or global communication between them. In Appendix I we empirically validate that having all of the steps is necessary for good expressiveness.

Let's consider a Simplified AgentNet version of this model, where the agent can only decide if it prefers transitions to explored nodes, unexplored nodes, going back to the previous node, or staying on the current one. Such an agent can already recognize cycles. If the agent always prefers unexplored nodes and marks every node it visits, which it can do uniquely as agents are uniquely identifiable, it will know once it has completed a cycle (Figure 1 (c)). Similarly, if for $c$ steps an agent always visits a new node, and it always sees all the previously visited nodes, it has observed a $c$-clique (Figure 1 (d)). This clique detection can be improved if the agent systematically visits all neighbors of a node $v$, going back to $v$ after each step, until a clique is found. This implies that even this simple model could distinguish the Rook's 4×4 and Shrikhande graphs, which are indistinguishable by 2-WL[1], but can be distinguished by detecting 4-cliques. If we have more than one agent they could successfully run such algorithms in parallel to improve the chances of recognizing interesting substructures.

## 4 THEORETICAL ANALYSIS

We first analyze the theoretical expressiveness of the AgentNet model described above. We begin with the case of a single agent, and then we move on to having multiple agents simultaneously. The proofs and more detailed discussion of our theorems are available in Appendices A and B.

### 4.1 EXPRESSIVENESS WITH A SINGLE AGENT

We first study the expressiveness of AgentNet with a single agent only. That is, assume that an agent is placed on the graph at a specific node $v$. In a theoretical sense, how much is an agent in our model able to explore and understand the region of the graph surrounding $v$?

Let us define the $r$-hop neighborhood of $v$ (denoted $N^r(v)$) as the subgraph induced by the nodes that are at distance at most $r$ from $v$. One fundamental observation is that AgentNets are powerful

---

[1]Note that there are two slightly different ways in the literature to index the WL hierarchy; here we use the so-called folklore variant, where 2-WL is already much more expressive than 1-WL. See e.g. [39] for details.

enough to efficiently explore and navigate the $r$-hop neighborhood around the starting node $v$. In particular, AgentNets can compute (and maintain) their current distance from $v$, recognize the nodes they have already visited, and ensure that they move to a desired neighbor in each step; this already allows them to efficiently traverse all the nodes in $N^r(v)$.

**Lemma 1.** *An AgentNet can learn to execute a iteratively deepening depth-first search of its $r$-hop neighborhood, visiting each node of $N^r(v)$ in $\ell \leq O(r \cdot |N^r(v)|)$ steps altogether.*

We note that in many practical cases, e.g. if the nodes of interest for a specific application around $v$ can already be identified from their node features, then AgentNets can traverse the neighborhood even more efficiently with a depth-first search. We discuss this special case (together with the rest of the proofs) in more detail in Appendix A.

While using these methods to traverse the neighborhood around $v$, an AgentNet can also identify all the edges between any two visited nodes; intuitively speaking, this allows for a complete understanding of the neighborhood in question. More specifically, if the transition functions of the agent are implemented with a sufficiently expressive method (e.g. with multi-layer perceptrons), then we can develop a maximally expressive AgentNet implementation, i.e. where each transition function is injective, similarly to GIN in a standard GNN setting [75]. Together with Lemma 1, this provides a strong statement: it allows for an AgentNet that can distinguish any pair of different $r$-hop neighborhoods around $v$ (for any finite $r$ and $\Delta$, where $\Delta$ denotes the maximal degree in the graph).

**Theorem 2.** *There exists an injective implementation of an AgentNet (with $\ell \leq O(r \cdot |N^r(v)|)$ steps) which computes a different final embedding for every non-isomorphic $r$-hop neighborhood that can occur around a node $v$.*

This already implies that with a sufficient number of steps, running an AgentNets from node $v$ can be strictly more expressive than a standard GNN from $v$ with $r$ layers, or in fact any GNN extension that is unable to distinguish every pair of non-isomorphic $r$-hop neighborhoods around $v$. In particular, the Rook's 4×4 and Shrikhande graphs are fundamental examples that are challenging to distinguish even for some sophisticated GNN extensions; an AgentNet can learn to distinguish these two graphs in as little as $\ell = 11$ steps. Since the comparison of the theoretical expressiveness of GNN variants is a heavily studied topic, we add these observations as an explicit theorem.

**Corollary 3.** *The AgentNet approach can also distinguish graphs that are not separable by standard GNNs or even more powerful GNN extensions such as PPGN [51], GSN [11] or DropGNN [59].*

Intuitively, Theorem 2 also means that AgentNets are expressive enough to learn any property that is a deterministic function of the $r$-hop neighborhood around $v$. In particular, for any subgraph $H$ that is contained within distance $r$ of one of its nodes $v_0$, there is an AgentNet that can compute (in $O(r \cdot |N^r(v)|)$ steps) the number of occurrences of $H$ around a node $v$ of $G$, i.e. the number of times $H$ appears as an induced subgraph of $G$ such that node $v$ takes the role of $v_0$.

**Lemma 4.** *Let $H$ be any subgraph of radius at most $r$ around a specific node $v_0$. Then there exists an AgentNet that can compute the number of occurrences of $H$ around a specific node $v$ of $G$.*

For several applications, cliques and cycles are often mentioned as some of the most relevant substructures in practice. As such, we also include more specialized lemmas that consider these two structures in particular. In these lemmas, we say that an event happens with high probability (*w.h.p*) if an agent can ensure that it happens with probability $p$ for an arbitrarily high constant $p < 1$.

**Lemma 5.** *There exists an AgentNet that can count cliques (of any size) in $\ell = 2 \cdot \Delta - 1$ steps, but there is no AgentNet that can count them w.h.p. in less steps.*

**Lemma 6.** *There exists an AgentNet that can count $c$-cycles in $\ell = \Theta(r \cdot |N^r(v)|)$ steps, but there is no AgentNet that can count them w.h.p. in less than $2 \cdot |N^r(v)| - \lfloor \frac{c}{2} \rfloor$ steps.*

In general, these theorems will allow us to show that if a specific subgraph appears much more frequently in some graph $G_1$ than in another graph $G_2$, then we can already distinguish the two graphs with only a few agents (we formalize this in Theorem 9 for the multi-agent setting).

Note that all of our theorems so far rely on the agents recognizing the structures around their starting point $v$. This is a natural approach, since in the general case, without prior knowledge of the structure of the graph, the agents cannot make more sophisticated (global) decisions regarding the directions in which they should move in order to e.g. find a specific structure.

However, another natural idea is to consider the case when the agents do not learn an intelligent transition strategy at all, but instead keep selecting from their neighbors *uniformly at random*. We do not study this case in detail, since the topic of such random walks in graphs has already been studied exhaustively from a theoretical perspective. In contrast to this, the main goal of our paper is to study agents that learn to make more sophisticated navigation decisions. In particular, a clever initialization of the Simplified AgentNet transition function already ensures in the beginning that the movement of the agent is more of a conscious exploration strategy than a uniform random walk.

Nonetheless, we point out that many of the known results on random walks automatically generalize to the AgentNets setting. For example, previous works defined a *random walk access* model [17; 20] where an algorithm begins from a seed vertex $v$ and has very limited access to the graph: it can (i) move to a uniform random neighbor, (ii) query the degree of the current node, and (iii) query whether two already discovered nodes are adjacent. Several algorithms have been studied in this model, e.g. for counting triangles in the graph efficiently [7]. Since an AgentNet can execute all of these fundamental steps, it is also able to simulate any of the algorithms developed for this model.

**Theorem 7.** *An AgentNet can simulate any algorithm developed in the random walk access model.*

## 4.2 Multiple agents

We now analyze the expressiveness of AgentNet with multiple agents. Note that the main motivation for using multiple agents concurrently is that it allows for a significant amount of parallelization, hence making the approach much more efficient in practice. Nonetheless, having multiple agents also comes with some advantages (and drawbacks) in terms of expressive power.

We first note that adding more agents can never reduce the theoretical expressiveness of the AgentNet framework; intuitively speaking, with unique IDs, the agents are expressive enough to disentangle their own information (i.e. the markings they leave at specific nodes) from that of other agents.

**Lemma 8.** *Given an upper bound $b$ on agent IDs, there is an AgentNet implementation that always computes the same final embedding starting from $v$, regardless of the actions of the remaining agents.*

This shows that even if we have multiple agents, they always have the option to operate independently from each other if desired. Together with Theorem 2, this allows us to show that if two graphs differ significantly in the frequency of some subgraph, then they can already be distinguished by constantly many agents. More specifically, let $G_1$ and $G_2$ be two graphs on $n$ nodes, let $H$ be a subgraph that can be traversed in $\ell$ steps as discussed above, and let $\gamma_H(G_i)$ denote the number of nodes in $G_i$ that are incident to at least one induced subgraph $H$.

**Theorem 9.** *Let $G_1$ and $G_2$ be graphs such that $\gamma_H(G_1) - \gamma_H(G_2) \geq \delta \cdot n$ for some constant $\delta > 0$. Then already with $k \in O(1)$ agents an AgentNet can distinguish the two graphs w.h.p.*

In general, Lemma 8 shows that if the number of steps $\ell$ is fixed, then we strictly increase the expressiveness by adding more agents. However, another (in some sense more reasonable) approach is to compare two AgentNet settings with the same number of total steps: that is, if we consider an AgentNet with $k$ distinct agents, each running for $\ell$ steps, then is this more expressive than an AgentNet with a single agent that runs for $k \cdot \ell$ steps?

We point out that in contrast to Lemmas 1-6 (which hold in every graph), our results for this question always describe a specific graph construction, i.e. they state that we can embed a subgraph $H$ in a specific kind of graph $G$ such that it can be recognized by a given AgentNet configuration, but not by another one. A more detailed discussion of the theorems is available in Appendix B.

On the one hand, it is easy to find an example where having a single agent with $k \cdot \ell$ steps is more beneficial: if we want to recognize a very large structure, e.g. a path on more than $\ell$ nodes, then this might be straightforward in a single-agent case, but not possible in the case of multiple agents.

**Lemma 10.** *There is a subgraph $H$ (of radius larger than $\ell$) that can be recognized by a single agent with $k \cdot \ell$ steps, but not by $k$ distinct agents running for $\ell$ steps each.*

On the other hand, it is also easy to find an example where the multi-agent case is more powerful at distinguishing two graphs, e.g. if this requires us to combine information from two distant parts of the graph that are more than $k \cdot \ell$ steps away from each other.

**Lemma 11.** *There is a pair of non-isomorphic graphs (of radius larger than $k \cdot \ell$) that can be distinguished by $k$ distinct agents with $\ell$ steps each, but not by a single agent with $k \cdot \ell$ steps.*

However, neither of these lemmas cover the simplest (and most interesting) case when we want to find and recognize a single substructure $H$ of small size, i.e. that has a radius of at most $\ell$. In this case, one might first think that the single-agent setting is more powerful, since it can essentially simulate the multi-agent case by consecutively following the path of each agent. For example, if two agents $a_1$, $a_2$ meet at a node $v$ while identifying a structure in the multi-agent setting, then a single agent could first traverse the path of $a_1$, then move back to $v$ and traverse the path of $a_2$ from here.

However, it turns out that this is not the case in general: one can design a construction where a subgraph $H$ can be efficiently explored by multiple agents, but not by a single agent. Intuitively, the main idea is to develop a *one-way tree* structure that can only be efficiently traversed in one direction: in the correct direction, the next step of a path is clearly distinguishable from the node features, but when traversing it the opposite way, there are many identical-looking neighbors in each step.

**Theorem 12.** *There exists a subgraph $H$ that can be recognized w.h.p. by $2$ agents in $\ell$ steps, but it cannot be recognized by $1$ agent in $c \cdot \ell$ steps (for any constant $c$).*

Finally, we note that as an edge case, AgentNets can trivially simulate traditional message-passing GNNs with $n$ agents if a separate agent is placed on each node, and performs the node update and neighborhood aggregation steps, while never transitioning to another node. However, if the problem at hand requires such an approach, it is likely more reasonable to use a traditional GNN instead.

## 5 EXPERIMENTS

In this section, we aim to demonstrate that AgentNet is able to recognize various subgraphs, correctly classifies large graphs if class-defining subgraphs are common, performs comparably on real-world tasks to other expressive models which are much more computationally expensive, and can also successfully solve said tasks better than GIN while performing much fewer than $n$ steps on the graph.

In all of the experiments, we parameterize all of the AgentNet update functions with 2-layer MLPs with a skip connection from the input to the output. For details on model implementation see Appendix C. Unless stated otherwise, to be comparable in complexity to standard GNNs (e.g. GIN) we use the number of agents equal to the mean number of nodes $n$ in the graphs of a given dataset. We further discuss the choice of the number of agents in Appendix J. For details on the exact setup of all the experiments and the hyperparameters considered see Appendix D.

### 5.1 SYNTHETIC DATASETS

First, to ensure that AgentNet can indeed recognize substructures in challenging scenarios we test it on synthetic GNN expressiveness benchmarks (Table 1). There we consider three AgentNet versions. The random walk AgentNet, where the exploration is purely random. The Simplified AgentNet, where each agent only predicts the logits for staying on the current node, going to the previous node, or going to any unexplored or any explored node, and finally the full AgentNet, which uses dot-product attention for transitions. Notice that the random walk AgentNet is not able to solve the 2-WL task, as a random walk is a comparatively limited and sample-intensive exploration strategy. The other two AgentNet models successfully solve all tasks, which means they can detect cycles and cliques. Since the simplified exploration strategy does not account for neighboring node features, in the rest of the experiments we will only consider the full AgentNet model.

Secondly, to test if AgentNet is indeed able to separate two graphs when the defining substructure is prevalent in the graph we perform the ablation in Figure 2a. We see that indeed in such case AgentNet can successfully differentiate them, even when observing just a fraction of the graph ($n >> k \cdot \ell$).

| Model | 4-CYCLES [59] | CIRCULAR SKIP LINKS [15] | 2-WL |
|---|---|---|---|
| GIN [75] | 50.0 ±0.0 | 10.0 ±0.0 | 50.0 ±0.0 |
| GIN with random features [64; 1] | 99.7 ±0.4 | 95.8 ±2.1 | 92.4 ±1.6 |
| SMP [71] | **100.0 ±0.0** | **100.0 ±0.0** | 50.0 ±0.0 |
| DROPGIN [59] | **100.0 ±0.0** | **100.0 ±0.0** | **100.0 ±0.0** |
| ESAN [8] | **100.0 ±0.0** | **100.0 ±0.0** | **100.0 ±0.0*** |
| 1-2-3 GNN [53] | **100.0 ±0.0** | **100.0 ±0.0** | **100.0 ±0.0†** |
| PPGN [51] | **100.0 ±0.0** | **100.0 ±0.0** | 50.0 ±0.0 |
| CRAWL [67] | **100.0 ±0.0** | **100.0 ±0.0** | **100.0 ±0.0** |
| RANDOM WALK AGENTNET | **100.0 ±0.0** | **100.0 ±0.0** | 50.5 ±4.5 |
| SIMPLIFIED AGENTNET | **100.0 ±0.0** | **100.0 ±0.0** | **100.0 ±0.0** |
| AGENTNET | **100.0 ±0.0** | **100.0 ±0.0** | **100.0 ±0.0** |

Table 1: Evaluation on datasets unsolvable by 1-WL (GIN). Regularized 4-cycles [59] dataset tests if a given graph has a 4 cycle, while Circular Skip Links dataset [15] asks to classify the graph by its cycle length. 2-WL dataset contains Rook $4 \times 4$ and Shrikhande graphs which are both co-spectral and indistinguishable by 2-WL, but the Rook graph has 4-cliques, while the Shrikhande graph does not. We highlight the best test scores in bold. AgentNet is able to detect both cycles and cliques. * ESAN can only solve the 2-WL dataset with the edge deletion policy, but not the other three policies. † Interestingly enough, while 1-2-3 GNN should have a similar expressive power to the 2-WL test, it solves this task. This is likely due to the fact that while all possible triplets are needed to simulate a 2-WL test, in practice only a subset of them is considered to reduce computational complexity [53].

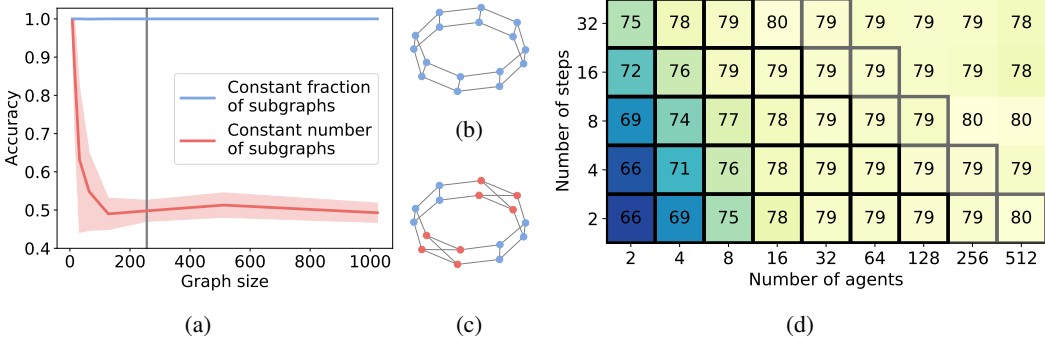

(a)  (b)  (c)  (d)

Figure 2: AgentNet sublinearity studies. First, we create a synthetic dataset with a variable density of the subgraphs of interest (a). The dataset contains ladder graphs (b) and ladder graphs with some cells replaced by crosses (c). The task is to differentiate the two graphs. We always use $k = 16$ agents, $\ell = 16$ steps, and gradually increase the ladder graph size from 16 nodes to 1024 nodes. We either preserve the crossed-cell density of $0.5$ (blue line) or we always have only two crossed cells in the ladder graph, independent of its size (red line). If the subgraph of interest is common in the graph (blue line) AgentNet successfully learns to differentiate the graphs, even when $k \cdot \ell < n$ (gray line). Secondly, to test how this transfers to real-world tasks we test AgentNet on a large protein DD dataset using a different number of agents and steps (d). Grey-bordered cells mark configurations that perform fewer node visits than GIN ($k \cdot \ell < 4 \cdot n$) and black-bordered cells mark configurations that perform less than $n$ node visits ($k \cdot \ell < n$). GIN achieved $77\%$ accuracy on this task. AgentNet outperforms it, even when using just $\frac{n}{8}$ node visits and matches GIN while using only $\frac{n}{16}$ node visits.

## 5.2 REAL-WORLD GRAPH CLASSIFICATION DATASETS

To verify that this novel architecture works well on real-world graph classification datasets, following Papp et al. [59] we use three bioinformatics datasets (MUTAG, PTC, PROTEINS) and two social network datasets (IMDB-BINARY and IMDB-MULTI) [76]. As graphs in these datasets only have 10s of nodes per graph on average, we also include one large-graph protein dataset (DD) [23] and one large-graph social dataset (REDDIT-BINARY) [76]. In these datasets, graphs have hundreds of nodes on average and the largest graphs have thousands of nodes (Appendix E). We compare our model to GIN, which has maximal 1-WL expressive power achievable by a standard message-passing GNN [75], and more expressive GNN architectures, which do not require pre-computed features [59; 8; 53; 51; 67]. As you can see in Table 2 our novel approach usually outperforms at least half

| Model | Complexity | MUTAG | PTC | PROTEINS | IMDB-B | IMDB-M | DD | RDT-B |
|---|---|---|---|---|---|---|---|---|
| GIN [75] | $O(n \cdot \ell)$ | 89.4 ±5.6 | 64.6 ±7.0 | 76.2 ±2.8 | 75.1 ±5.1 | 52.3 ±2.8 | 76.9 ±3.7 | 92.4 ±2.5 |
| DropGIN [59] | $O(r \cdot n \cdot \ell)$ | 90.4 ±7.0 | 66.3 ±8.6 | 76.3 ±6.1 | 75.7 ±4.2 | 51.4 ±2.8 | 76.4 ±3.4 | 89.9 ±1.7 |
| ESAN [8]* | $O(r \cdot n \cdot \ell)$ | 91.1 ±7.0 | **69.2 ±6.5** | 77.1 ±4.6 | **77.1 ±2.6** | **53.5 ±3.4** | **81.2 ±2.3** | 93.3 ±1.3 |
| 1-2-3 GNN [53]† | $O(n^4 \cdot \ell)$ | 88.8 ±7.0 | 64.0 ±6.0 | 76.8 ±3.7 | 73.6 ±2.2 | 51.1 ±3.8 | OOM | OOM |
| PPGN [51]* | $O(n^3 \cdot \ell)$ | 90.6 ±8.7 | 66.2 ±6.5 | **77.2 ±4.7** | 73 ±5.8 | 50.5 ±3.6 | OOM | OOM |
| CRAWL [67] | $O((n + k \cdot m) \cdot \ell)$ | 90.4 ±7.1 | 68.0 ±6.5 | 76.2 ±3.7 | 73.4 ±2.1 | 47.8 ±3.9 | 78.3 ±5.5 | 92.8 ±2.2 |
| 1-AGENTNET | $O(\ell)$ | 89.4 ±10.9 | 66.6 ±7.6 | 75.1 ±3.4 | 74.9 ±3.9 | 52.3 ±3.9 | 67.4 ±3.0 | 77.9 ±3.0 |
| AGENTNET | $O(k \cdot \ell)$ | **93.6 ±8.6** | 67.4 ±5.9 | 76.7 ±3.2 | 75.2 ±4.6 | 52.2 ±3.8 | 80.1 ±2.7 | **94.2 ±1.2** |
| Rank | | 1st | 3rd | 4th | 3rd | 3rd | 2nd | 1st |

Table 2: Graph classification accuracy (%). DropGIN and ESAN models use $r \approx O(n)$ versions of each graph, which makes them of quadratic complexity in practice. We set $k \approx n$ for AgentNet models to have a comparable setting to GIN. We also compare it to a 1-AgentNet, which uses only one agent and cannot visit the whole graph. The state-of-the-art random walk model CRAWL computes $k = n$ random walks, that are $m = 50$ steps long at every layer. * We report the best result achieved by any of the different model versions. PPGN has 3 different versions and ESAN has 8 different versions. † Originally 1-2-3 GNN used a slightly different evaluation setup. We re-trained it to follow the same experimental setup as the other baselines [75] (see Appendix D). All other results come from the respective papers or were trained using the original setup and code.

of the more expressive baselines. It also compares favorably to CRAWL [67], the state-of-the-art random-walk-based model in 6 out of the 7 tasks. Even the AgentNet model that uses just a single agent matches the performance of GIN, even though in this case the agent cannot visit the whole graph, as for these experiments we only consider $\ell \in \{8, 16\}$ (Appendix D) and using only one agent reduces expressiveness. Naturally, 1-AgnetNet performance deteriorates on the large graphs, while AgentNet does very well. The higher-order methods cannot even train on these large graphs on a 24GB GPU. To train the ESAN and DropGIN models on them we have to only use $5\%$ of the usual $n$ different versions of the same graph [8]. For DropGIN, this results in a loss of accuracy. While ESAN performs well in this scenario, it requires lengthy pre-processing and around 68GB to store the pre-processed dataset on disk (original datasets are $< 150$MB), which can become prohibitive if we need to test even larger graphs or larger datasets (Appendix H).

To test in detail how AgentNet performance depends on the number of agents and steps in large real-world graphs we perform an ablation on the DD dataset (Figure 2d). We see that many configurations perform well, even when just a fraction of the graph is explored. Especially when fewer than $n$ agents are used ($n \approx 284$).

The previous datasets do not use edge features. As mentioned in Section 3, this is a straightforward extension (Appendix C). We test this extension on the OGB-MolHIV molecule classification dataset, which uses edge features [38]. In Table 3 we can see that AgentNet performs well in this scenario. In Appendix

| | OGB-MolHIV | |
|---|---|---|
| Model | Validation | Test |
| GIN [75] | 82.32 ±0.90 | 75.58 ±1.40 |
| GIN + virtual node [75] | 84.79 ±0.68 | 77.07 ±1.49 |
| CRAWL [67] | 83.44 ±0.96 | 77.78 ±0.87 |
| ESAN [8]* | 84.28 ±0.90 | 78.00 ±1.42 |
| AGENTNET | 84.77 ±0.92 | **78.33 ±0.69** |

Table 3: Test and validation ROC-AUC (%) on the OGB-MolHIV dataset with edge features. *Best result achieved by any version.

F and H you can see how the model performs on even more tasks. In Appendix K we also show that AgentNet indeed can make use of the defining subgraphs in real-world datasets.

## 6 CONCLUSION

In this work, we presented a novel AgentNet architecture for graph-level tasks. We provide an extensive theoretical analysis, which shows that this architecture is able to distinguish various substructures that are impossible to distinguish with traditional GNNs. We show that AgentNet also brings improvements to real-world datasets. Furthermore, if features necessary to determine the graph class are frequent in the graph, AgentNet allows for classification in sublinear or even a constant number of rounds. To our knowledge, this is the first fully differentiable GNN computational model capable of graph classification which inherently has this feature, without requiring explicit graph sparsification – it learns which substructures are worth exploring on its own.

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

# A  PROOFS FOR SECTION 4.1

## A.1  INJECTIVE IMPLEMENTATION

The fundamental idea behind developing a maximally expressive AgentNet implementation is to ensure that the functions learned by the agent are injective. A similar proof technique for developing injective GNN implementations has already been applied to analyze the expressive power of standard GNNs [75] and also some GNN extensions [59]; we refer the reader to these papers for more technical details on this proof approach. As in the rest of these proofs, we assume that the range of the node features is a finite set, and we also have a finite upper bound on the maximal degree $\Delta$ of the graph. With an induction, this already shows that after any finite number of time steps $t$, the range of possible embeddings (of agents or nodes) is still finite.

The main tool we will use is the following: assume we have a finite number of embeddings $h_0, ..., h_{j-1} \in \mathbb{R}$, and assume for simplicity that they are from the interval $(0, 1)$. Then there exists a function $f : (0,1)^j \to (0,1)$ that is injective on its entire domain.; that is, if we have $\hat{h}_1, ..., \hat{h}_j$ such that $h_i \neq \hat{h}_i$ for at least one $i \in \{0, ..., j-1\}$, then $f(h_0, ..., h_{j-1}) \neq f(\hat{h}_0, ..., \hat{h}_{j-1})$. One possible way to construct such an $f$ is as follows: we consider the binary representation of each $h_i$, and encode it in the bits of $f(h_0, ..., h_{j-1})$ at positions that are equal to $i$ modulo $j$. That is, if $i = z_1 \cdot j + z_2$ for some $0 \leq z_2 < j$, then we define the $i$-th bit of $f(h_0, ..., h_{j-1})$ (after the decimal point) to be equal to the $z_1$-th bit of $h_{z_2}$ (after the decimal point). Note that this function is indeed injective, since for any $f(h_0, ..., h_{j-1})$, we can uniquely reconstruct all the values $h_0, ..., h_{j-1}$. Note that the numbers are only restricted to the interval $(0, 1)$ for the sake of simplicity; if $h_i > 1$, then we can encode the bits of $h_i$ by alternatingly moving in both directions from the decimal point.

As such, for any finite number of values, there exists an injective function $f$ that combines these into a single real embedding. Such a function $f$ can be approximated arbitrarily closely with a multi-layer perceptron according to the universal approximation theorem, and hence an AgentNet can also have the theoretical expressiveness to learn any such function.

From here, the concrete design of the injective AgentNet implementation only requires the repeated application of this idea. As a first step, we need to create an injective node update function $f_v$ for the single agent case; for this, we encode the values $v_i^{t-1}$ and $a^{t-1}$ (and for convenience, also the value of $t$) with the method above. That is, whenever we have either $v_i^{t-1} \neq \hat{v}_i^{t-1}$ or $a^{t-1} \neq \hat{a}^{t-1}$ or $t \neq \hat{t}$, then our update function will ensure $f_v(v_i^{t-1}, a^{t-1}) \neq f_v(\hat{v}_i^{\hat{t}-1}, \hat{a}^{\hat{t}-1})$. As described above, such a function $f_v$ exists, and hence can be learned by a sufficiently powerful AgentNet.

As a next step, the node essentially executes a message-passing phase around its neighborhood; this can again be done injectively. In particular, the work of [75] describes how to design an injective multiset function, i.e. an aggregation of neighbors that returns a different embedding for every possible multiset of neighboring embeddings. We can once again combine this with the embedding $v_i^t$ of the node as discussed above, creating an $f_n$ that is injective in $v_i^t$ and the multiset $v_j^t \in N(v_i)$.

In the single-agent case, we can simply select $f_a(a_i^{t-1}) = v_i^t$, since this already contains all the information available to the agent at this point.

Intuitively, the injective implementation means that any decision that can be made by a deterministic algorithm from the same information can also be executed in our setting. In particular, this also holds for the node transition functions: in each step, we can select the transition function $f_p$ to model almost any categorical distribution $\pi$ over the closed neighborhood $\{v_0\} \cup N(v_0)$ of the current node $v_0$, based on the current node state $a_{v_0}^t$. That is, given the desired probabilities $\pi(v_j)$, inverting the softmax function determines the set of required logits, and we need to define $f_p$ accordingly. Recall that in the injective implementation, $a_{v_0}^t$ already determines both $v_0^t$ and the multiset of $v_j^t$ in $N(v_0)$; hence when defining $f_p(a_i^t, v_j^t)$, we can indeed already determine the set of desired logits from $a_i^t$ alone, and assign the appropriate value for each $v_j^t$.

For a categorical distribution $\pi$ to be obtainable as a transition function this way, it needs to satisfy two simple properties: (i) $\pi(u) \in (0, 1)$ for any $u \in \{v_0\} \cup N(v_0)$, and (ii) if $u_1, u_2$ has the same current embedding, then $\pi(u_1) = \pi(u_2)$. Furthermore, property (i) is also not strict in the sense that if a probability of 0 is desired, then we can also select a sufficiently small transition value such that the

probability is essentially $0$ after the softmax; that is, if $\ell$ and $\Delta$ are both finite, then we can always set $\pi(u)$ small enough to ensure that the probability of executing any of these "0-probability transitions" at any point during the entire $\ell$ steps is below an arbitrary constant $\epsilon$. As such, the only real restriction on the set of obtainable categorical distributions is that two nodes with the same embedding must receive the same transition probabilities. Since an injective agent ensures that every visited node has a different embedding, this only means the following restriction in practice: if $v_0$ has multiple unvisited neighbors with the same initial node features, then the transition probabilities to these nodes will be identical. If any categorical distribution over the closed neighborhood satisfies this property, then there exists an $f_p$ function corresponding to this distribution, and an agent can learn to approximate this.

For completeness, we also introduce a technical modification to our injective agent implementation to ensure that the range of the function $f_v$ can never accidentally coincide with any of the (finite) possible initial values of the node features. To do this, we encode one more finite value in the representation of $f_v$ (in every $j$-th bit, as before). Since there are only finitely many possible node features, we can e.g. use the $i$-th of these extra bits to ensure that that $f_v$ is not equal to the $i$-th possible initial feature (this approach is also known as the "diagonalization method"). This ensures that any node that is not yet visited will have a different embedding than all the already visited nodes.

Finally, let us make some simple observations about this injective implementation that can serve as building blocks for our more complex algorithms later. First of all, we note that each node encodes the entire history of the agent from all of the time steps when the node was visited; in particular, a node is aware of each time unit when it was visited by the agent. Furthermore, in each step, an agent can uniquely determine the neighbor of the current node that it has arrived from. Also, for each neighbor of the current node, an agent can determine whether it has already been visited or not.

## A.2 GRAPH TRAVERSAL METHODS

With the injective implementation discussed above, agents can already learn to intelligently traverse the $r$-hop neighborhood of their starting node $v$ in the graph. We first discuss the iterative depth-first search approach that provides Lemma 1, and then also comment on the simpler case when a depth-first search is sufficient.

*Proof of Lemma 1.* Since the agent is not aware of the size and shape of its $r$-hop neighborhood in advance, it needs to execute an iteratively deepening depth-first search (IDDFS) in the neighborhood to ensure that it discovers all nodes from $N^r(v)$, but only these nodes. The IDDFS algorithm iterates a depth limit $d$ from 1 to $r$, and in each iteration, it executes a depth-first search from $v$ up to depth $d$ only. This ensures that all the nodes at distance $d$ from $v$ are already identified in iteration $d$, but no nodes that are farther away from $v$ are visited. Note that we cannot achieve the same with a regular depth-first search with depth limit $r$: it might happen that some parts of the search tree from node $u$ are discarded due to this depth limit, and we only find a shorter path to $u$ later which shows that the distance from $v$ to these discarded nodes is, in fact, smaller than $r$.

It is easy to see that an injective AgentNet (as described above) can indeed execute such an IDDFS algorithm; we just need to select the transition function $f_p$ appropriately. Note that the current embedding $a_j$ of the agent already uniquely determines the current iteration $d$ that the agent is executing; alternatively, we can also save this current value $d$ on separate bits when constructing the injective functions of the agent (recall that we can encode any finite number of values with the same approach). This also implies that the current embedding of each node $u$ determines whether $u$ was already visited in iteration $d$ or not. Besides this, the embedding of each (already visited) node $u$ determines the distance of $u$ from $v$: in the iteratively deepening setting, this is simply the index of the first iteration where this node was visited. Finally, when the agent is on a node $u$, it can determine the predecessor of $u$ in the depth-first search tree of the current iteration: it can consider the first time step $t$ when $u$ was visited in iteration $d$, and the predecessor of $u$ is the node that was visited in time step $(t-1)$.

Based on these observations, we can describe the transition function $f_p$ with the following few simple rules. When the agent is at distance at most $(d-1)$ from $v$, and there are still neighbors that have not been visited in the current iteration, then we assign a large constant value of $c$ to these nodes in $f_p$, and a value of (essentially) $0$ to all other nodes. When the agent is at distance $d$ from $v$, or all of its

neighbors have already been visited in iteration $d$, then the agent moves backward: we set a value of $c$ for the predecessor of the current node, and a value of $0$ to all other nodes. Finally, if the agent is at $v$ and all of its neighbors have been visited in iteration $d$ already, then it begins iteration $(d+1)$ and sets the value of all neighbors to $c$ (it selects the next neighbor uniformly at random).

Any iteration of the IDDFS traverses a depth-first search tree with at most $N^r(v)$ nodes; each edge of the tree is traversed at most twice, resulting in at most $2 \cdot |N^r(v)|$ steps. Since the number of iterations is $r$, our agent indeed needs $\ell \leq O(r \cdot |N^r(v)|)$. We note that in most practical cases, we can even drop the factor $r$ from this expression: e.g. when the graph region around $v$ is a $\Delta$-regular tree, then we will have most nodes in $N^r(v)$ at distance exactly $r$ from $v$, and thus all iterations except for the last one will become asymptotically irrelevant. $\qquad\square$

This iteratively deepening search method is only required because we always need to be aware of the distance of the current node from $v$ in order to decide whether we need to explore the graph further in a specific direction. However, in some special cases, we can apply a much more efficient depth-first search (DFS) traversal of the neighborhood; this removes the factor $r$ from the required number of steps.

One simple example of such a setting is when the connected component of the graph containing $v$ is very small. In particular, assume that the entire component only consists of $n_0$ nodes; while this might be unusual in actual applications, it is relatively frequent in the synthetic datasets that are often used to measure the expressive power of different GNN extensions. For another example, consider applications where the nodes that are important for our purpose have some specific features that make them easy to distinguish from other nodes; in this case, an injective AgentNet can learn to only consider these nodes while processing the graph (i.e. set the transition probability to all other nodes to $0$). As such, we can fictitiously remove every from the graph that does not have the appropriate features, and if the connected component of $v$ in the remaining graph (the "interesting part" of the region around $v$) is relatively small with only $n_0$ nodes, then we can again restrict our exploration strategy to this subgraph.

In the cases mentioned above, the agent can traverse the entire subgraph of size $n_0$ around $v$ with a depth-first search approach: it becomes unnecessary to maintain the distances from $v$ anymore since the agent traverses the entire connected component anyway. This is a much more efficient method in terms of the number of steps, not requiring an extra factor $r$: the entire connected component around $v$ can be traversed in $2 \cdot n_0 - 3$ steps (or in terms of the radius $r$ of the connected component, in $2 \cdot |N^r(v)| - 3$ steps). This is because the DFS tree has $n_0 - 1$ edges, each of these is traversed at most twice, and the edges leading to the last discovered node are traversed only once (but for this we can only subtract a single edge in the worst case).

We note that this bound is indeed tight, i.e. even in this DFS-based setting, we cannot explore every neighborhood in $\ell = 2 \cdot n_0 - 4$ steps. This also allows us to show that there are subgraphs $H$ of radius $r$ such that no AgentNet can decide w.h.p. in $2 \cdot n_0 - 4$ steps whether $H$ occurs as an induced substructure around $v$. A concrete example is shown for this in the proof of the negative result in Lemma 5: the neighborhood of interest (that can contain a triangle) in this case consists of $n_0 = 1 + \Delta = 4$ nodes, but it is not possible w.h.p. in $2 \cdot n_0 - 4 = 4$ steps to decide whether $v$ has an incident triangle.

## A.3 Recognizing structures

Given an agent that systematically traverses its neighborhood, we can now consider the claims of recognizing specific substructures.

We begin with the proof of Theorem 2. We point out that in the context of graphs with node features, we when we say that two graphs $G_1(V_1, E_1)$ and $G_2(V_2, E_2)$ are isomorphic, then besides the regular graph-theoretic definition of having $\sigma : V_1 \to V_2$ such that $(v_1, v_2) \in E_1 \iff (\sigma(v_1), \sigma(v_2)) \in E_2$, we also require that $v_1$ and $\sigma(v_1)$ have identical features for all $v_1 \in V_1$.

*Proof of Theorem 2.* According to Lemma 1, there exists an AgentNet that explores every node in the $r$-hop neighborhood of $v$; furthermore, during this traversal, the agent also becomes aware of the features of every node in $N^r(v)$, and all edges between pairs of nodes in $N^r(v)$ when the second endpoint of the edge is visited. This allows an agent to uniquely identify the entire $r$-hop

neighborhood around $v$. More specifically, if an injective agent produces the same output embedding for two neighborhoods, then this implies that for the two traversals $T_1, T_2$, the following properties must all hold: (i) the nodes visited in the $t$-th and $t'$-th steps of $T_1$ are the same node if and only if the nodes visited in the $t$-th and $t'$-th steps of $T_2$ are the same node, (ii) the $t$-th visited nodes of $T_1$ and $T_2$ have the same node features, and (iii) the $t$-th and $t'$-th visited nodes of $T_1$ are adjacent if and only if the $t$-th and $t'$-th visited nodes of $T_2$ are adjacent. These properties provide a clear bijection between the nodes of the two neighborhoods, also preserving node features and edges; this implies that the two neighborhoods are isomorphic.

This shows that an injective AgentNet always assigns a different embedding to non-isomorphic neighborhoods. However, we also need to ensure that isomorphic neighborhoods, on the other hand, obtain the same embedding. Indeed, with the injective AgentNet described so far, it could easily happen that two nodes have isomorphic neighborhoods, but an agent traverses these neighborhoods in a different order, and hence computes a different embedding in the two cases.

In order to resolve this, we only need to observe that there is a function assigning every possible IDDFS traversal to the isomorphism class of $N^r(v)$, and a sufficiently powerful agent can learn to apply this function on the final embedding. More specifically, let $\mathcal{G}$ denote the set of all different (non-isomorphic) graphs of radius at most $r$ (and degree at most $\Delta$) around $v$. We have already seen that if two $r$-hop neighborhoods are non-isomorphic, then our injective agent always produces a different final embedding for them. This implies that the final embedding uniquely determines the graph induced by $N^r(v)$, i.e. there exists a well-defined function $\psi_1 : \mathbb{R} \to \mathcal{G}$ which assigns the graph induced by $N^r(v)$ to every possible final embedding generated by the agent. Finally, consider a function $\psi_2 : \mathcal{G} \to \mathbb{R}$ which assigns the numbers $\{1, \ldots, |\mathcal{G}|\}$ to the graphs in $\mathcal{G}$ in arbitrary order. Then $\psi := \psi_2 \circ \psi_1$ is simply a function $\psi : \mathbb{R} \to \mathbb{R}$, and according to the universal approximation theorem, it can be learnt e.g. by an MLP implementation. Applying this function $\psi$ on the final embedding (in the last step of the traversal) ensures that the final output of the agent describes the isomorphism class of $N^r(v)$, and hence two starting nodes receive the same final embedding if and only if their $r$-hop neighborhoods are isomorphic. $\qquad\square$

*Proof of Corollary 3.* Corollary 3 follows easily from Theorem 2 and the fact that there are known limitations to the expressiveness of each of the listed GNN extensions. In the case of a standard GNN, two small cycles of different lengths are already a simple example of indistinguishability [75]. For the 2-WL algorithm (and hence equivalently, PPGN), the Rook 4×4 and Shrikhande graphs are a well-known example on 16 nodes that are not distinguishable. For GSN and DropGNN, the analysis of [58] describes an example construction that cannot be distinguished without preprocessing structures of size $\Theta(\Delta)$ or removing $\Theta(\Delta)$ nodes. Hence for any fixed parametrization of GSN or DropGNN (preprocessing substructures of fixed size, or removing a fixed number of nodes), there exists a pair of neighborhoods that cannot be distinguished by GSN or DropGNN. These example graphs have $\Delta + 1$ nodes, so they can still be distinguished by AgentNet in $\ell = 2 \cdot \Delta - 1$ steps. $\quad\square$

Lemma 4 follows easily from Theorem 2.

*Proof of Lemma 4.* The $r$-hop neighborhood around $v$ already determines the number of occurrences of any structure $H$ of radius at most $r$ from $v$. That is, if we denote the set of all possible non-isomorphic $r$-hop neighborhoods around $v$ by $\mathcal{N}^r$, then there is a deterministic function $f_H : \mathcal{N}^r \to \mathbb{N}$ that describes the number of occurrences of $H$ for each neighborhood in $\mathcal{N}^r$. Theorem 2 shows that we can learn an injective function $f_A : \mathcal{N}^r \to \mathbb{R}$ that describes the final embedding of an agent; hence we can define the function $g = f_H \circ f_A^{-1}$ that assigns the appropriate number of occurrences of $H$ to any final embedding. This function $g$ can also be learned according to the universal approximation theorem, and hence an AgentNet can learn to execute this function in its last step. $\quad\square$

We discuss these substructure-related claims for cliques and cycles explicitly, which are both important substructures for specific applications. Note that the positive parts of the lemmas are deterministic, i.e. given a specific number of steps, there is an agent implementation that always (with probability 1) returns the desired number of subgraphs as its final embedding; meanwhile, the negative parts claim that if the number of steps is small, then no agent can return the correct number w.h.p.

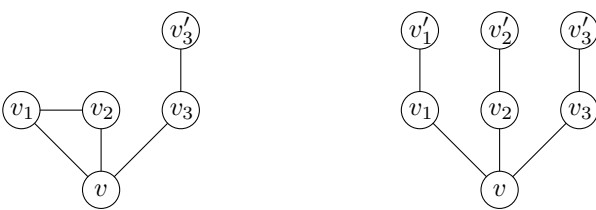

Figure 3: Example graphs $G_1$ (left) and $G_2$ (right) for the negative result in Lemma 5.

*Proof of Lemma 5.* As a special case of the IDDFS discussed before (with $r = 1$), an injective AgentNet can learn to mark the starting node $v$, then transition to an unvisited neighbor of $v$ in each odd step, and then move back to $v$ in even steps. This allows it to visit all neighbors of $v$ in $2 \cdot \Delta - 1$ steps, and identify the entire 1-hop neighborhood of $v$ as discussed in Lemma 1. Since every clique containing $v$ is completely included in this induced subgraph, this AgentNet can compute the number of cliques incident to $v$ for any clique size.

On the other hand, if $\ell \leq 2 \cdot \Delta - 2$, then the agent does not have enough steps to visit every neighbor of $v$; as such, it might not detect a clique if the last unvisited neighbor of $v$ is contained in it. For a concrete example, consider two graphs where $v$ has degree 3 (and also $\Delta = 3$); in $G_1$, we have edges $\{(v, v_1), (v, v_2), (v, v_3), (v_1, v_2), (v_3, v_3')\}$ , i.e. a triangle and a path of length 2 incident to $v$, while in $G_2$, we have edges $\{(v, v_1), (v, v_2), (v, v_3), (v_1, v_1'), (v_2, v_2'), (v_3, v_3')\}$, i.e. three paths of length 2 incident to $v$; see Figure 3 for an illustration. Assume that all nodes begin with identical node features.

In the first step, any agent can only move to a uniform random neighbor $u$ of $v$ (staying at the current node is not a reasonable action in this setting). In either of the graphs, the agent observes the same situation after the first step, so it must move to the other neighbor of $u$ with a fixed probability $p_1$, and back to $v$ with probability $(1 - p_1)$. Note that if $p_1 \geq \frac{1}{2}$, and if the current node $u$ happens to be $v_3$ (this has probability $\frac{1}{3}$), then the agent cannot distinguish the two graphs in the remaining steps (with probability at least $\frac{1}{6}$).

If the agent returns to $v$ in the second step, then in the remaining two steps, it gains the highest possible amount of information by visiting another neighbor $u'$ of $v$ and then moving to the other neighbor of $u'$. However, once again, if $u = v_3$ (this has probability $\frac{1}{3}$), then the agent cannot distinguish the two graphs from the path it has traversed. Thus $(1 - p_1) \geq \frac{1}{2}$ implies a failure probability of at least $\frac{1}{6}$ again. This shows for any choice of $p_1$, the agent fails to distinguish the two graphs with an arbitrarily high constant probability (i.e. it higher than $\frac{5}{6}$). $\qquad\square$

*Proof of Lemma 6.* A cycle of $c$ nodes has radius $\lfloor \frac{c}{2} \rfloor$, so according to Lemma 4, an AgentNet can indeed count the number of incident cycles of length $c$ in $O(c \cdot |N^{\lfloor \frac{c}{2} \rfloor}(v)|)$ steps.

On the other hand, if the $\lfloor \frac{c}{2} \rfloor$-hop neighborhood of $v$ consists only of paths of length $\lfloor \frac{c}{2} \rfloor$ starting from $v$, and we need to visit each of these paths to see if the endpoints of two paths are adjacent (thus forming a $c$-cycle), then $\ell = 2 \cdot |N^{\lfloor \frac{c}{2} \rfloor}(v)| - \lfloor \frac{c}{2} \rfloor - 2$ steps can indeed be required: intuitively, we need to visit each path starting from $v$ up to a distance $\lfloor \frac{c}{2} \rfloor$, and then return to $v$ on each occasion except for the last one. For a concrete example, we can consider the graphs $G_1, G_2$ from the proof of Lemma 5 again: a triangle is a cycle for $c = 3$, we have $2 \cdot |N^{\lfloor \frac{c}{2} \rfloor}(v)| - \lfloor \frac{c}{2} \rfloor - 2 = 2 \cdot 4 - 1 - 2 = 5$ in this case, and we have already seen that that no AgentNet can distinguish these graphs w.h.p. in less than 5 steps.

We note that for a tight lower bound in this case, we would also need to incorporate a further factor $r = \lfloor \frac{c}{2} \rfloor$ (to account for the iterative phases of the IDDFS); however, this would require a much more complex construction, since here intuitively we would also need to ensure that leaving out any of the iterative phases would result in an incorrect traversal of the neighborhood. $\qquad\square$

### A.4 RANDOM WALK ACCESS MODEL

Finally, we have already outlined the proof of Theorem 7 in Section 4.1.

*Proof of Theorem 7.* The injective version of AgentNet can easily carry out all three fundamental steps of the random walk access model. Moving to a uniform random neighbor in each step can be implemented by a simple transition function $f_p$ that assigns the same value to each neighbor. Besides this, the agent ensures (by encoding the time step $t$) that it leaves a unique node embedding at each of the visited nodes, so an injective AgentNet can recognize any of these nodes (or in fact, any possible subset of these nodes) in all subsequent steps. In this case, an injective AgentNet can ensure in general that in the $i$-th step (for any finite $i$), it computes an embedding that is injective in the (i) degree of the current node and (ii) set of previously visited nodes that are adjacent to the current node. That is, in the injective implementation, after any number of steps, the agent is aware of the degree sequence observed so far and the adjacency relations between all the already discovered nodes. Such an agent is already in possession of all the information that can be queried in the random walk access model. As such, the output of any algorithm in this model can be expressed as a deterministic function of the final embedding of the agent, and hence due to the universal approximation theorem, an agent can approximate this output if implemented by a sufficiently expressive method (e.g. multi-layer perceptron). $\square$

# B    PROOFS FOR SECTION 4.2

## B.1    SIMPLER CLAIMS IN THE MULTI-AGENT SETTING

We now discuss the theorems on expressiveness with multiple agents. We begin with Lemma 8, which only requires a further extension of the injective implementation provided earlier.

*Proof of Lemma 8.* The lemma assumes that each agent has a unique ID from 1 to some known upper bound $b$ (alternatively, we can also select the agent IDs from a predetermined finite set). In this case, we can modify the injective construction described in Appendix A to ensure that the $i$-th agent only uses the bits at positions $i$ modulo $b$ for its own encoding. This means that when summing the embeddings of different agents in $f_v$, we again have an injective function, even if all the agents visit the same node at the same time. As such, any single agent can determine its final embedding based on only its own part of the node embeddings, and disregard the embeddings of other agents. $\square$

After this proof, Theorem 9 only requires a few more steps. Note that the rest of the claims in this section discuss the ability of AgentNets (with a specific $k$ and $\ell$) to distinguish two graphs $G_1$ and $G_2$. To formalize this concept, we say that an AgentNet *can distinguish* $G_1$ and $G_2$ if it returns a different output (in the final readout function) for the two graphs w.h.p., i.e. it returns a given value $\alpha_1$ w.h.p. in case of $G_1$, and another value $\alpha_2$ w.h.p. in case of $G_2$. We say that it cannot distinguish two graphs if this is not possible, i.e. if there is a constant upper bound on the success probability.

*Proof of Theorem 9.* From Lemmas 4 and 8, it follows that there exists an AgentNet implementation such that if an agent is placed on a node $v$, then it computes a final embedding of 1 if $v$ is incident to a copy of $H$ in $G$, and 0 otherwise (note that we can include it in the last transition function to also convert the injective outputs into these more convenient 0-1 values). Consider an aggregation of the agents that simply sums up their final embeddings; this results in a final embedding that equals to the number of starting nodes $v$ that were incident to a copy of $H$.

Let $\gamma_i = \frac{\gamma_H(G_i)}{n}$ for $i \in \{1, 2\}$, and $\gamma' = \frac{1}{2} \cdot (\gamma_1 - \gamma_2)$. Recall that $\gamma_1 - \gamma' \geq \frac{\delta}{2}$ and $\gamma' - \gamma_2 \geq \frac{\delta}{2}$. The starting point of the agents is chosen uniformly at random and independently from each other; hence if we run an AgentNet with $k$ agents in $G_i$, then each of these will output 1 with probability $\gamma_i$ and 0 otherwise, and thus our final embedding (after aggregating the agents) will follow a binomial distribution $X$ with parameters $k$ and $\gamma_i$. We can then use a Chernoff bound to upper bound the probability that the sum falls below (above) $\gamma' \cdot k$ in $G_1$ ($G_2$, respectively). If the value is above $\gamma' \cdot k$ in $G_1$ and below $\gamma' \cdot k$ in $G_2$ w.h.p., then a simple classification function that compares the sum of embeddings to this value $\gamma' \cdot k$ can already distinguish the two graphs.

Let $p$ denote the constant probability we require for our definition of w.h.p. Let $\epsilon := \frac{\delta}{2 \cdot \gamma_i}$ (for a fixed $i \in \{1, 2\}$); then for the expected value $\mathbb{E}X$ of the binomial distribution, we have $\epsilon \cdot \mathbb{E}X =$

$\frac{\delta}{2 \cdot \gamma_i} \cdot \gamma_i \cdot k = \frac{\delta}{2} \cdot k$. According to the Chernoff bound, we have

$$\Pr\left(|X - \mathbb{E}X| \geq \epsilon \cdot \mathbb{E}X\right) \leq 2 \cdot e^{-\frac{1}{3} \cdot \epsilon^2 \cdot \mathbb{E}X} = 2 \cdot e^{-\frac{1}{3} \cdot \frac{\delta^2}{4 \cdot \gamma_i^2} \cdot \gamma_i \cdot k}.$$

While this looks like a complicated expression, $\frac{\delta^2}{4 \cdot \gamma_i^2} \cdot \gamma_i$ is simply a constant in our case. This implies that for any choice of $p$, there exists a high enough constant value $k$ such that the expression on the right-hand side is smaller than $(1 - p)$. By our choice of $\epsilon$, having $|X - \mathbb{E}X| < \epsilon \cdot \mathbb{E}X$ implies $X > \gamma'$ in $G_1$ and $X < \gamma'$ in $G_2$. This means that the probability of having $X \leq \gamma'$ in $G_1$ is at most $(1 - p)$, and similarly, the probability of having $X \geq \gamma'$ in $G_2$ is at most $(1 - p)$. □

We next present the proofs of Lemmas 10 and 11 that compare the single-agent and multi-agent settings.

*Proof of Lemma 10.* A simple example for such a subgraph $H$ is a path on $\ell$ nodes; let us assume for simplicity that the nodes have a single feature, and the value of this feature is 0 for all nodes of the path.

Consider a graph $G_1$ that consists of such a path $H$, and $n - \ell$ further nodes (with feature 1) that are all connected to the first node of the path. If $n$ is large enough (and $k$ is smaller, e.g. a constant), then w.h.p. all of the agents will start on a node with feature 1. As such, in $\ell$ steps, neither of them is able to reach the other end of the path.

This means that if we consider another graph $G_2$ where the only difference is that the last node of the path also has feature 0, then any AgentNet with $k$ agents fails to distinguish the graphs w.h.p. In contrast to this, a single agent with at least $\ell + 1$ steps can easily learn to walk to the other end of the path (by simply moving to the neighbor with feature 1 in the first step, and then always to the unmarked neighbor of feature 1), and hence distinguish the two graphs.

Note that if we want a version of this construction that has $\Delta = O(1)$, we can simply replace the nodes added to the beginning of the path by a complete $(\Delta - 1)$-ary tree of depth $h$: the root node is the first node of the path, and every node (apart from the leaves) has $(\Delta - 1)$ children. By assigning a different feature to each level of the tree, we can ensure that wherever an agent begins in the tree, it can learn to directly walk to the root in at most $h$ steps (we discuss these one-way trees in detail later). Note that we have $h \leq O(\log n)$ in this graph. Hence if we select $\ell = h$, then we can again ensure w.h.p. that all nodes start in the tree, and hence cannot reach the end of the path after $\ell$ steps. In contrast to this, with $k \cdot \ell$ steps (assuming $k \geq 2$), a single agent can always reach the end of the path and hence distinguish the two graphs. □

*Proof of Lemma 11.* Let both $G_1$ and $G_2$ consist of two identical connected components of equal size; in the first component, both graphs have a node feature of 0 on all nodes, whereas, in the second component, the node features are 0 and 1 in $G_1$ and $G_2$, respectively. If having multiple components are undesired, we can also connect the two components with a path of length $k \cdot \ell + 1$ (with node features of, say, 2).

In the case of a single agent, the agent appears in the first part of the graph with probability (almost) $\frac{1}{2}$ and is hence unable to distinguish the two graphs w.h.p., regardless of its actions.

However, in the case of a sufficiently high number of agents $k$ (i.e. if $1 - 2^k > p$ is satisfied, and assuming that the connecting path is an asymptotically irrelevant part of the graph), at least one of the agents begins in the second component; in this case, they can easily recognize a node with feature 1, and hence distinguish the two graphs. □

## B.2 PROOF OF THEOREM 12

It remains to prove the more involved Theorem 12 regarding the comparison of the two settings. For completeness, we first state the theorem in a more precise form:

**Theorem 12 (detailed).** *There exists a pair of non-isomorphic graphs $G_1$, $G_2$ and a connected subgraph $H$ such that*

- *$H$ appears as a subgraph in $G_1$, but not in $G_2$,*

- *there exists an AgentNet with $2$ agents and $\ell$ steps that can distinguish the two graphs w.h.p. by walking through the nodes of $H$ in $G_1$,*

- *there exists no AgentNet with $1$ agent and $c \cdot \ell$ steps (for any constant $c$) that can distinguish the two graphs w.h.p.*

That is, our proof here is a more surprising construction than that of Lemma 11: the whole graph has a diameter $2 \cdot \ell$ only, and there is a specific structure $H$ that distinguishes the two graphs, but a single agent is unable to navigate this structure. We first present the key idea to develop such constructions, which we call one-way trees.

**Definition 13.** *A one-way tree is a complete $(\Delta - 1)$-ary tree of depth $h$, i.e. a rooted tree where every node (apart from the leaves) has exactly $(\Delta - 1)$ children. The single node feature of every node in the $i$-th level is the integer $i$ (for $i \in \{1, ..., h\}$).*

Note that the node degrees are still bounded by $\Delta$ this way. The key idea of the tree is that an AgentNet can easily lean to walk towards the root in this tree, but finding a specific leaf is not possible without traversing a significant portion of the tree.

**Lemma 14.** *Consider a one-way tree, select one of its leaf nodes $u$, and let us add an extra neighbor (with feature $(h + 1)$) to this leaf $u$.*

- *There exists an AgentNet implementation which, if placed on $u$ initially, can reach the root of the tree in $\ell = h$ steps.*

- *There exists no AgentNet implementation which, if placed on the root node initially, can reach $u$ in $\ell = \frac{1}{2} \cdot (\Delta - 1)^h$ steps with probability larger than $\frac{1}{2}$.*

*Proof.* If an agent is placed at $u$ initially, then in each step, it can easily distinguish the parent node of its current node in the tree, since this is the only node that has a smaller feature. That is if the agent learns to transition to the node with feature $h$ in the first step, with feature $(h - 1)$ in the second step, and so on, then it can reach the root of the tree in $h$ steps without ambiguity (i.e. with probability 1).

On the other hand, when starting from the root, the agent has no way to distinguish the different children of the current node from each other. The agent can only proceed by visiting each of the $(\Delta - 1)^h$ leaves of the tree sequentially to find the leaf which has a neighbor with feature $(h + 1)$. In $\frac{1}{2} \cdot (\Delta - 1)^h$ steps, the agent can visit strictly less than half of the leaves, so it will only find $u$ with probability less than $\frac{1}{2}$. $\qquad\square$

With this tool, we can already move on to the proof of Theorem 12.

*Proof of Theorem 12.* Let $b$ be a constant parameter (to be discussed later), and consider a node $v$ with feature 0. Let us consider $b$ distinct one-way trees of depth $h_1$ (we will call these *primary trees*), and connect the root of all these trees to $v$. So far we do this in both of our graphs. Then in the case of $G_2$, let us further select one of the leaf nodes $v_0$ in one of the primary trees, and attach the root node of another one-way tree of depth $h_2$ (called the *secondary tree*) to $v_0$. In contrast to this, in $G_1$, we select an arbitrary leaf node in *every* primary tree, and we attach another one-way tree of depth $h_2$ (a secondary tree) to each of these leaf nodes.

For our substructure $H$, let us select a path on $2 \cdot h_1 + 3$ nodes, such that the node features on the path (in order) are $(1, h_1, h_1 - 1, ..., 2, 1, 0, 1, 2, ..., h_1 - 1, h_1, 1)$. Note that this structure appears in $G_1$: we start from a root node of a secondary tree, then move to the attached leaf of the primary tree, then to the root of the primary tree, then to $v$, then to the root of another primary tree, then all the way down to another leaf node that has a secondary tree attached, and finally to the root of this secondary tree. On the other hand, $H$ does not appear as a substructure in $G_2$, since $v_0$ is the only node in the graph which has feature $h_1$ and has a neighbor with feature 1.

We select $h_2$ significantly larger than $h_1$; with this, we can ensure that with arbitrarily high probability, the agents are always placed in one of the secondary trees (even in $G_2$, since $b$ is only a constant). Furthermore, note that if we have $k = 2$ agents, then with probability $1 - \frac{1}{b}$ they are placed into different secondary trees initially in $G_1$. That is, by selecting the constant $b$ large enough, we can

ensure that the two agents are placed into two distinct secondary trees with a probability of at least $p$ (i.e. w.h.p.). Finally, we select $\ell = h_2 + h_1$.

In the case of 2 agents, the agents can learn to always move to the root of the trees as discussed in the proof of Lemma 14: first to the root of the secondary tree, and then to the root of the primary tree. In $\ell$ steps, both agents can reach $v$. In the case of $G_1$, if the two agents begin in different secondary trees, then they only meet at node $v$; in $G_2$, they will always already meet at the root of the single secondary tree in the graph. Since the agents can learn to check if they share their location with the other agent in a given step, they can easily distinguish these two cases. Hence they can separate the two graphs w.h.p.: in $G_1$ they are correct with probability at least $1 - \frac{1}{b} \geq p$, and in $G_2$ they are correct with probability 1.

However, a single agent is unable to do the same. It can walk to node $v$ in $\ell$ steps in either of the graphs, but from there it cannot decide if there exists another secondary tree attached to the graph in one of the other primary trees. More specifically, according to the second part of Lemma 14, the agent cannot find another secondary tree with probability larger than $\frac{1}{2}$ in $\frac{1}{2} \cdot (\Delta - 1)^{h_1}$ steps, so if $\frac{1}{2} \cdot (\Delta - 1)^{h_1} > c \cdot \ell$ (for any given $c$ and $\Delta \geq 3$, we can ensure this when choosing the value of $h_1$), then the agent only finds a copy of $H$ with probability $\frac{1}{2}$ at most, even in $G_1$ where such a subgraph exists. Hence the agent encounters this situation (not finding another secondary tree) with a probability of at least $\frac{1}{2}$ in both graphs. This implies that it cannot distinguish the two graphs w.h.p.: whichever classification ($G_1$ or $G_2$) the agent assigns to this situation, it is wrong in one of the two graphs with a probability of at least $\frac{1}{2}$. $\qquad\square$

Note that if we want an example for Theorem 12 where the two graphs have the same number of nodes (e.g. to extend it to the case when the agents are aware of $n$), we need to adjust it slightly. We can define $G_1'$ to consist of $b$ independent copies of our original $G_1$. Now both graphs contain $b$ secondary trees. Since $h_2$ is significantly larger than $h_1$, their number of nodes is essentially equal: that is, we can add a small independent component $G_2^+$ to our original $G_2$ in order to ensure that $G_1'$ and $G_2' := G_2 \cup G_2^+$ have the same size (i.e. $G_2^+$ is just an arbitrary graph on $(b-1) \cdot (1 + b \cdot \sum_{i=0}^{h_1 - 1} (\Delta - 1)^i)$ nodes). By choosing $h_2$ much larger than $h_1$, we can ensure that $G_2^+$ is an asymptotically irrelevant part of $G_2'$, i.e. it holds with an arbitrarily high constant probability that neither of the agents is placed within $G_2^+$. Now given these graphs $G_1'$ and $G_2'$, a single agent is still unable to distinguish the two cases, since they observe the same situation in both graphs with a probability of at least $\frac{1}{2}$. However, when we have two agents, in $G_2'$ they meet w.h.p. only at $v$, whereas in $G_1'$, they always meet either already at the root of the secondary tree, or not at all; hence we can distinguish the two graphs.

## C    MODEL IMPLEMENTATION

In this section, we will discuss all aspects of the practical implementation of our model and its possible extensions. The model is implemented using PyTorch [60] and PyTorch Geometric [29][2].

### C.1    GENERAL NOTES AND INITIALIZATION

We use 2-layer MLPs to parameterize all of the model update functions (node update $f_v$, neighborhood aggregation $f_n$, agent update $f_a$) and the different inputs are concatenated. We use a Leaky ReLU [49] activation function with a negative slope of $0.01$. As you will see below, we use skip connections for both node and agent embeddings. This necessitates the use of Layer Normalization (LN) for the MLP inputs. This makes each MLP a Pre-LN residual block [74]. Both agents and nodes have $h$-dimensional embedding vectors.

To ensure starting node embeddings are of the correct dimension and that their sum is injective we first pre-process all node features with an MLP. Since for the majority of the graphs we consider we expect the agents to observe a large fraction of the graph, we process all nodes once in parallel. To stick to the strictly sub-linear setting the model can be modified to only process node embeddings the first time they are observed.

---

[2]Code is available at `https://github.com/KarolisMart/AgentNet`

As we need our agents to be uniquely identifiable, we use $k$ learnable $h$-dimensional embedding vectors, which are used as the starting agent state. We initially place agents uniformly at random on the graph.

## C.2 MODEL STEPS AND POSSIBLE EXTENSIONS

Let's now see, how the four model steps performed upon visiting a node we described in Section 3 are implemented in practice. Note that the same functions (MLPs) are used for every agent. However, because agents have different embeddings, the functions can produce different outcomes for different agents.

Here, we will also show how the model can be extended with edge features and global communication between the agents. In practice, we always include global agent communication. If the given dataset does not make use of edge features, we use a simplified model where the pre-aggregation update functions $\phi_{N(v)\to v}$ and $\phi_{a\to v}$ are set to identity. As MLPs are universal function approximators, previous MLP can already directly approximate $\phi_{N(v)\to v}$ and $\phi_{a\to v}$, thus we do not lose expressiveness by excluding them [75]. One could also imagine, that providing the agents with the total number of nodes in the graph could help. If the task, for example, is to estimate the density of triangles in the graph, by observing just a fraction of it. Such conditioning can be trivially achieved by including $n$ as input to every update function. However, in our experiments, we did not consider such augmentation.

**Node update.** As discussed earlier we use a skip connection. If we have edge features, then $\phi_{a\to v}$ includes both the agent state and the edge $e_{a_j}$ it took to arrive at the current node $\phi_{a\to v}\left(a_j^{t-1}, e_{a_j}\right) =$ LeakyReLU $\left(\text{LinearLayer}\left(\text{LN}\left(a_j^{t-1}, e_{a_j}\right)\right)\right)$. In this case, we use a negative slope of $0.2$. If there are no edge features $\phi_{a\to v}\left(a_j^{t-1}, e_{a_j}\right) = a_j^{t-1}$:

$$v_i^t = v_i^{t-1} + f_v\left(v_i^{t-1}, \sum_{a_j^{t-1}\in A(v_i)} \phi_{a\to v}\left(a_j^{t-1}, e_{a_j}\right)\right) \quad \textbf{if } |A(v_i)| > 0 \textbf{ else } v_i^{t-1}.$$

If we want to have global information between the agents, we can include mean of their embeddings $\frac{1}{k}\sum_{a_j\in A} a_j$ as another input to $f_v$. As all of the subsequent operations make use of the current node state $v_i^t$, this global information can impact every other update function. Note that since $k$ is fixed, this mean is also injective.

To ensure all of these sum pooling operations do not cause value explosion and training problems but retain their expressiveness we implement them as mean scaled by the log of summand count. This is also true for the neighbor aggregation.

**Neighborhood aggregation.** In the same fashion, a skip connection is used, while $\phi_{N(v)\to v}$ is either identity or a linear layer followed by a non-linearity, depending on whether the graph has edge features or not. $e_{v_j\to v_i}$ is used to denote an edge from node $v_j$ to node $v_i$. This results in a GIN-like convolution [75] with a skip connection:

$$v_i^t = v_i^t + f_n\left(v_i^t, \sum_{v_j^t\in N(v_i)} \phi_{N(v)\to v}\left(v_j^t, e_{v_j\to v_i}\right)\right) \quad \textbf{if } |A(v_i)| > 0 \textbf{ else } v_i^t.$$

**Agent update.** Similarly, the agent update is straightforward and can take into account the edge $e_{a_i}$ the agent used to reach the current node if edge features are used:

$$a_i^t = a_i^{t-1} + f_a\left(a_i^{t-1}, v_{V(a_i)}^t, e_{a_i}\right).$$

**Agent transition.** First, let us consider the Simplified AgentNet. In this case, for each agent $a_i$ we track which nodes $v_j$ have been explored by it $x(a_i, v_j) \in [0, 1]$. Every time step we decay the values $x = 0.9 \cdot x$ and set the values for the current nodes $V(a_i)$ of all of the agents $a_i \in A$ to $x(a_i, V(a_i)) = 1$. We can now use these exploration values $x$ to construct the simplified transition

function $f_{ps}$. Using agent embedding an MLP $g(a_i^t)$ produces four logits $[g_p, g_c, g_e, g_u] = g(a_i^t)$, respectively for the previous node, the current node, explored nodes and unexplored nodes. Then, for each neighboring node, we determine its final logits as a weighted sum of these four values. To check if a given node $v_j$ was the agent's $a_i$ previous node $V^{t-1}(a_i)$ we use an indicator variable $\mathbb{1}_{v_j = V^{t-1}(a_i)}$. Another indicator variable checks if the node is the agent's current node $\mathbb{1}_{v_j = V(a_i)}$. The explored and unexplored node logits $g_e$ and $g_u$ are interpolated using the node's exploration value $x(a_i, v_j)$:

$$f_{ps}\left(a_i^t, v_j^t\right) = g_p(a_i^t) \cdot \mathbb{1}_{v_j = V^{t-1}(a_i)} + g_c(a_i^t) \cdot \mathbb{1}_{v_j = V(a_i)} + g_e(a_i^t) \cdot x(a_i, v_j) + g_u(a_i^t) \cdot (1 - x(a_i, v_j)),$$

$$z_{a_i \to v_j} = f_{ps}\left(a_i^t, v_j^t\right) \quad \text{for} \quad v_j^t \in N^t(a_i).$$

To ensure efficient training in the beginning we want as many agents as possible to observe the defining subgraphs. This requires exploration. Unfortunately, random walks are sample inefficient and require many steps to even walk a whole simple connected component e.g. a cycle. It is known, that just preventing the random walk from backtracking already greatly increases the sample efficiency [43]. In the same vein, we initialize the learnable bias of the $g(a_i^t)$ output layer such that $[g_p, g_c, g_e, g_u] \approx [0, -1, 0, 5]$ and the model is initially biased to focus on the yet unexplored nodes. Note that in principle one can also restrict $g(a_i^t)$ to be just a set of learnable global bias parameters that do not depend on the agent to produce an even simpler model.

For the full AgentNet, we use dot-product attention $f_p$ to determine the next node, where the query vector $Q(a_i^t)$ is a linear projection of the agent embedding and the key vector $K(v_i^t, v_j^t, e_{v_j \to v_i})$ is a linear projection of the source node embedding, target node embedding, and any edge features.

$$f_p\left(a_i^t, v_j^t\right) = \frac{Q(a_i^t)^T K(v_i^t, v_j^t, e_{v_j \to v_i})}{\sqrt{h}}$$

To still ensure that the exploration is efficient at the beginning of the training, we combine dot-product attention with values produced by the Simplified AgentNet transition function $f_{ps}$. In this case, the transition logits $g_p$, $g_c$, $g_e$, and $g_u$ are just global learnable bias parameters that are independent of the agent and are initialized the same way as before:

$$z_{a_i \to v_j} = f_{ps}\left(a_i^t, v_j^t\right) + f_p\left(a_i^t, v_j^t\right) \quad \text{for} \quad v_j^t \in N^t(a_i).$$

The resulting logits $z_{a_i \to v_j}$ are used to sample a node from the neighborhood using straight-through Gumbel Softmax:

$$V(a_i) \leftarrow \text{GumbelSoftmax}\left(\left\{z_{a_i \to v_j} \quad \text{for} \quad v_j^t \in N^t(a_i)\right\}\right).$$

To ensure the gradients flow through the Gumbel Softmax sampling, we interpret its output as a sparse one-hot vector (where only the 1s are present). We use the resulting agent $\to$ node adjacency matrix for agent pooling in the node update step and for selecting the appropriate node in the agent update step, thus multiplying the 1s carrying the gradients with the correct embedding vectors.

## C.3 READOUT

As we have stated in Section 3, agent embeddings are pooled together to make the final graph-level decision. In practice, following Xu et al. [75] we pool agent embeddings after each agent update step and sum the individual step predictions to make the final prediction:

$$o_{a_i}^t = \phi_o\left(a_i^t\right),$$

$$\bar{o} = \sum_t f_o\left(\frac{1}{k} \sum_{a_i \in A} o_{a_i}^t, \max\left\{o_{a_i}^t \quad \text{for} \quad a_i \in A\right\}\right).$$

Here $\phi_o$ is a 2-layer MLP used to project agent embeddings before the pooling. We use both mean and max pooling because in theory there could be two kinds of problems, respectively: problems where agents collectively need to decide how commonplace certain features are and problems where it is sufficient that just one agent finds a class-defining feature. As the final readout $f_o$ we use a simple linear layer.

## C.4    TIME-STEP CONDITIONING

Technically, the agents are able to track time themselves, as the model is injective and they act every time step. However, to make the task easier for the model, we condition all of the update functions (node update $f_v$, neighborhood aggregation $f_n$, agent update $f_a$) and the readout function $\phi_o$ on the current time step. We achieve this by adding the Transformer sinusoidal position (time step) embedding [68] to the inputs of each function (MLP).

## C.5    POSSIBLE MODEL SIMPLIFICATIONS

We implemented the model to be as flexible as possible and to match the theoretical analysis. However, the resulting model performs quite a few operations every step. We could simplify it at a loss of some expressiveness. For example, node update and neighborhood aggregation steps can be merged, but this will cause the neighborhood aggregation to see the neighbors as they were in the previous time step. Similarly, the agent update step could be merged into this unified update step, if we are content with producing the same delta update for every agent that is on the same node. Even the agent transition probabilities could be incorporated in this unified step if for example we would model the node aggregation after the GATv2 [12] convolution and would use the final attention head to produce transition probabilities for every edge. The same transition probabilities would then be used by every agent. Naturally, this would reduce the model's expressiveness, but it would result in a simpler and fully parallelized model. However, in this work, we aimed to provide an expressive, theoretically motivated model and show that it works well in practice. We leave the investigation of various model simplifications for future work.

## D    EXPERIMENTAL SETUP

For AgentNet in all of the tasks, we use AdamW optimizer [47] with a weight decay of $0.1$. We set the initial learning rate to $10^{-4}$ and decay it over the whole training to $10^{-11}$ using a cosine schedule. We also clip the global gradient norm to $1$. In all of the cases, Gumbel-Softmax temperature is set to $\frac{2}{3}$ as this has been suggested as a robust choice when the distribution has only a few categories [50].

### D.1    SYNTHETIC DATASETS

**Expressiveness benchmarks.**    To ensure all of the baseline models as well as the different AgentNet versions have a fair shot at solving the given tasks we perform a generous grid search for all of them. We consider batch size $\in \{50, 300\}$ and hidden units $\in \{64, 128\}$ and learning rate $\in \{0.001, 0.0005, 0.0001\}$. For AgentNet we set number of agents $k = n$ (16 for 4-Cycles and 2-WL, 41 for Circular Skip Links) and consider $\ell \in \{16, 64\}$. For other GNN architectures, we also include $\ell \in \{4, 8\}$ as they tend to perform worse with high depth. On top of this for AgentNet we consider the number of agents $k \in \{2, n\}$ as these problems should be solvable even with few agents. In fact, we did observe that having $k = n$ agents can make the model convergence slower, and for 2-WL dataset results in mean accuracy of $98 \pm 2\%$. We train each configuration using 10 random seeds and report the mean and the standard deviation of the best configuration after 10 thousand training steps. For the baseline models, we use the same training setup as for AgentNet. The baselines themselves were chosen as the most expressive models among their class of expressive GNNs. PPGN [51] represents the higher-order GNNs and matches 2-WL in expressive power, GIN with random features [64; 1] represents non-equivariant node identification, SMP [71] represents the equivariant node identification scheme, while DropGNN [59] represents the models that use many different versions of a graph to make a prediction [59; 8; 19].

As described in Appendix C.2 the AgentNet and the Simplified AgentNet use initial attention weights biased for exploration. We also tested these models on these synthetic benchmarks without this bias and they still successfully solved the task. However, we kept this bias in the other experiments due to the theoretically better sample complexity.

**Subgraph density ablation.**    As discussed in Figure 2, we set $k = 16$, $\ell = 16$, use 128 hidden units and a batch size of 200 and train using 10 random seeds for 10 thousands steps. We report the mean and the standard deviation of accuracy at the end of the training. While such a large batch

size and a large number of hidden units are not necessary for this task, we aimed to make sure these hyperparameters are large enough to rule out any training or capacity issues in the results of this ablation study.

## D.2 REAL-WORLD GRAPH CLASSIFICATION DATASETS

**TU datasets.** We follow the evaluation setup by Xu et al. [75], as was done by all of the baseline models. We perform a 10-fold cross-validation and report the mean and the standard deviation. In line with baseline models, we train for 350 epochs, and for each of the datasets we perform a grid search over the batch size $\in \{32, 128\}$, hidden units $\in \{32, 64, 128\}$ and number of steps $\ell \in \{8, 16\}$. We always set the number of agents to the mean number of nodes in the graphs of the corresponding dataset (see Table 4). As REDDIT-BINARY has some very high-degree nodes with thousands of neighbors (Table 4) this can cause memory issues when many agents end up on the same high-degree node at the same time and compute their transition probabilities. To avoid this, for REDDIT-BINARY we use $k = 350$ agents instead of $430$ and consider $\ell \in \{4, 8\}$. For the DD dataset, we use the usual setup, as it does not have such high-degree nodes. Neither DropGNN [59] nor ESAN [8] or CRAWL [67] were trained on DD (DropGNN also wasn't trained on REDDIT-BINARY), thus we trained them using the original code[34] and hyperparameter tuning.

As authors of CRAWL [67] do not describe the hyperparameter search used, based on the hyperparameters they report in the paper, for CRAWL we perform a grid search over the batch size $\in \{10, 100\}$, hidden units $\in \{50, 100\}$ and dropout $\in \{0.0, 0.5\}$. We train all of the TU datasets not reported in the original paper for 350 epochs. Otherwise, the original code base[5] is unchanged.

We also re-trained 1-2-3 GNN [53] to follow the experimental setup by Xu et al. [75]. We use the original architecture[6] including the layer counts and a similar hyperparameter search as used for our model and by Xu et al. [75]. The search is performed over the batch size $\in \{32, 64\}$, the hidden unit count $\in \{16, 32, 64, 128\}$ and dropout ratio $\in \{0.0, 0.5\}$. The model is trained with Adam optimizer [42], a learning rate of $0.001$, and no weight decay as done originally [53]. We used a learning rate decay of $50\%$ every 50 epochs as done by Xu et al. [75]. We found this schedule to perform slightly better than the original decay on plateau used by Morris et al. [53].

**OGB.** We follow the standard evaluation setup proposed by Hu et al. [38]. Similarly to the previous graph classification tasks, we set $k$ to mean number of nodes ($n \approx 26$), considered number of steps $\ell \in \{8, 16\}$, batch size $\in \{32, 64\}$ and hidden units $\in \{64, 128\}$. We train the model with 10 random seeds for 100 epochs, select the model with the best validation ROC-AUC and report the mean and the standard deviation. The best setup proved to be using a batch size of $64$, $128$ hidden units, and $\ell = 16$ steps. In general, over most of the tasks we tested, we observed that a larger batch size tends to improve the AgentNet training and that having around 128 hidden units is a good choice, at least with our considered range of values for the number of agents $k$. When we train AgentNet on OGB-MolPCBA and OGB-PPA we use these best hyperparameters from OGB-MolHIV, only reducing the number of steps to 8 and setting $k = 150$ for OGB-PPA. Similarly, when we train CRAWL on OGB-MolHIV and OGB-PPA we use the setup used by the authors for OGB-MolPCBA [67]. When training ESAN on OGB-MolPCBA and OGB-PPA we also use the authors' setup for OGB-MolHIV [8].

## E GRAPH STATISTICS

In Table 4 we provide the graph statistics for all of the real-world datasets used in Section 5.2. As you can see, most of the commonly used TU datasets [54], QM9 [63], OGB-MolHIV and OGB-MolPCBA [38] have small graphs. The two large TU datasets we include (DD and REDDIT-BINARY) have much larger graphs, especially when considering the largest examples. REDDIT-BINARY also has some very high-degree nodes. To an extent, this is also true for the smaller social-graph datasets

---

[3]`https://github.com/KarolisMart/DropGNN`
[4]`https://github.com/beabevi/ESAN`
[5]`https://github.com/toenshoff/CRaWl`
[6]`https://github.com/chrsmrrs/k-gnn`

| Dataset | # graphs | Mean # nodes | Max # nodes | Min # nodes | Mean deg. | Max deg. |
|---|---|---|---|---|---|---|
| MUTAG | 188 | 17.9 | 28 | 10 | 2.2 | 4 |
| PTC | 344 | 25.6 | 109 | 2 | 2.0 | 4 |
| PROTEINS | 1113 | 39.1 | 620 | 4 | 3.73 | 25 |
| IMDB-B | 1000 | 19.8 | 136 | 12 | 9.8 | 135 |
| IMDB-M | 1500 | 13.0 | 89 | 7 | 10.1 | 88 |
| DD | 1178 | 284.3 | 5748 | 30 | 5.0 | 19 |
| RDT-B | 2000 | 429.6 | 3782 | 6 | 2.3 | 3062 |
| OGB-MolHIV | 41127 | 25.5 | 222 | 2 | 2.2 | 10 |
| OGB-MolPCBA | 437929 | 26.0 | 332 | 1 | 2.2 | 5 |
| OGB-PPA | 158100 | 243.4 | 300 | 50 | 18.62 | 299 |
| ZINC | 12000 | 23.2 | 37 | 9 | 2.2 | 4 |
| QM9 | 130831 | 18.0 | 29 | 3 | 2.0 | 5 |

Table 4: Graph statistics for the real-world datasets.

(IMDB-BINARY and IMDB-MULTI). OGB-PPA [38] included in Appendix H is also comprised of larger graphs on average, but lacks very large examples present in DD and REDDIT-BINARY.

## F PERFORMANCE ON POORLY ALIGNED TASKS

We want to check how our model performs in graph-level tasks it is not well aligned with. To this end, we test our AgentNet on the molecule property regression task on the QM9 dataset [63]. Conceptually, for this molecule property regression task, we probably want to perform message passing on each node every time, as we likely need to learn the exact geometric structure of the molecule (position of every node, relative to other nodes, taking charges and bonds into account). This would make AgentNet (with randomly placed agents) not well suited for this task. However, in Table 5 we can see that AgentNet still outperforms the non-expressive baselines (MPNN and 1-GNN) of similar computational complexity while performing comparably to the expressive baselines (1-2-3 GNN, PPGN, DropMPNN, and Drop-1-GNN) which have much higher computational complexity. This means that even on unfavorable tasks it can perform sufficiently well.

| Property | Unit | MPNN [34; 73] | 1-GNN [53] | 1-2-3 GNN [53] | PPGN [51] | DropMPNN [59] | Drop-1-GNN [59] | AgentNet |
|---|---|---|---|---|---|---|---|---|
| $\mu$ | Debye | 0.358 | 0.493 | 0.473 | 0.0934 | **0.059** | 0.453 | 0.254 |
| $\alpha$ | Bohr$^3$ | 0.89 | 0.78 | 0.27 | 0.318 | **0.173** | 0.767 | 0.198 |
| $\epsilon_{HOMO}$ | Hartree | 0.00541 | 0.00321 | 0.00337 | **0.00174** | 0.00193 | 0.00306 | 0.00183 |
| $\epsilon_{LUMO}$ | Hartree | 0.00623 | 0.00350 | 0.00351 | 0.0021 | 0.00177 | 0.00306 | **0.0016** |
| $\Delta\epsilon$ | Hartree | 0.0066 | 0.0049 | 0.0048 | 0.0029 | 0.00282 | 0.0046 | **0.0025** |
| $\langle R^2 \rangle$ | Bohr$^2$ | 28.5 | 34.1 | 22.9 | 3.78 | **0.392** | 30.8 | 1.28 |
| ZPVE | Hartree | 0.00216 | 0.00124 | 0.00019 | 0.000399 | **0.000112** | 0.000895 | 0.000232 |
| $U_0$ | Hartree | 2.05 | 2.32 | 0.0427 | **0.022** | 0.0409 | 1.80 | 0.145 |
| $U$ | Hartree | 2.0 | 2.08 | 0.111 | **0.0504** | 0.0536 | 1.86 | 0.146 |
| $H$ | Hartree | 2.02 | 2.23 | 0.0419 | **0.0294** | 0.0481 | 2.00 | 0.155 |
| $G$ | Hartree | 2.02 | 1.94 | **0.0469** | 0.24 | 0.0508 | 2.12 | 0.119 |
| $C_v$ | cal/(mol K) | 0.42 | 0.27 | 0.0944 | **0.0144** | 0.0596 | 0.259 | 0.0708 |

Table 5: Mean absolute errors on QM9 dataset [63]. The best-performing model is in bold.

For this task, we changed the neighborhood aggregation function $f_n$ to align better with the task, by using the continuous kernel-based convolutional operator proposed by Gilmer et al. [34] for the message pre-processing function $\phi_{N(v) \to v}$. This convolution is used by all of the baseline models we consider. Similarly, to stay close to baseline models we parameterize both the final readout $f_o$ and the $\phi_{a \to v}$ function used for agent aggregation in the node update by 2-layer MLPs. We also use the global information exchange between the agents, both in the node update step and in the agent update step. Otherwise, the training procedure is the same as for the other tasks.

## G CONVERGENCE WITH RANDOMNESS

The AgentNet model uses random agent placement, and initially when the model is untrained agent transitions are also random. This raises a natural question: how negatively does this randomness affect

AgentNet training? In Figure 4 we can see that the AgentNet always converges, and does this faster than an expressive GIN model, which also depends on randomness. This difference is particularly noticeable on harder tasks, such as distinguishing two 2-WL indistinguishable graphs.

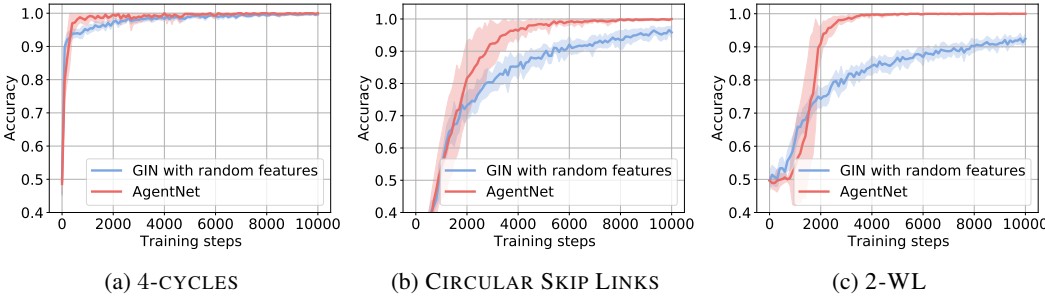

|  (a) 4-CYCLES  |  (b) CIRCULAR SKIP LINKS  |  (c) 2-WL  |

Figure 4: Convergence of test accuracy as a function of training steps for AgentNet and GIN with random features on hard synthetic expressiveness benchmarks. AgentNet always converges and does so faster than GIN with random features, especially on harder tasks, such as 2-WL. The curves correspond to the best set of hyperparameters from Table 1 experiments.

In this context, it is also worth noting, that for all of the OGB and TU dataset experiments we used the standard number of training epochs (100 for OGB and 350 for TU datasets). This means that the competitive results seen in Tables 2, 3 and 6 do not require extended training to counteract the randomness.

## H    ADDITIONAL REAL-WORLD EXPERIMENTS

In this section, we provide additional experiments on other graph classification tasks from Open Graph Benchmark [38]. OGB-MolPCBA dataset is a larger dataset for multi-class molecule classification and OGB-PPA is a classification dataset of ego-graphs from a protein-protein association network. These ego graphs are a bit larger and have around 243 nodes on average. In Table 6 you can see that AgentNet noticeably outperforms the GIN baselines, while on the small OGB-MolPCBA its performance is in-between GIN and GIN with a virtual node. The big gap in performance between GIN and GIN that uses a virtual which connects all other nodes, suggests that this task might depend on more global features. As we discussed in Appendix F we expect AgentNet to perform a bit worse on tasks where the global structure is more important than local structure.

| Model | OGB-MolPCBA (AP) | | OGB-PPA (Acc) | |
|---|---|---|---|---|
|  | Validation | Test | Validation | Test |
| GIN [75] | 23.05 ±0.27 | 22.66 ±0.28 | 65.62 ±1.07 | 68.92 ±1.00 |
| GIN + virtual node [75] | 27.98 ±0.25 | 27.03 ±0.23 | 66.78 ±1.05 | 70.37 ±1.07 |
| ESAN [8] | 28.46 ±0.22 | 26.64 ±0.27 | OOM | OOM |
| CRAWL [67] | 30.75 ±0.20 | **29.86 ±0.25** | 63.66 ±0.56 | 70.25 ±0.52 |
| AGENTNET | 26.22 ±0.26 | 25.49 ±0.27 | 67.24 ±0.65 | **72.33 ±0.62** |

Table 6: Test and validation average precision (%) on the OGB-MolPCBA dataset and accuracy (%) on OGB-PPA datasets. We were unable to train ESAN model on OGB-PPA dataset, as a machine with 256GB of main memory did not have enough RAM to pre-process the graphs for training (OOM). Note that the pre-processed graphs for the much smaller DD dataset (Table 4) already took 68GB of space on disk and required 120GB of main memory to load.

We also test our model on a popular ZINC-12K molecular graph regression dataset [24; 40], where the models need to predict the constrained solubility of the molecule. While this property is more defined by the global structure of the molecule, so might not align very well with AgentNet inductive bias (Appendix F), some local features, like cycle counts are important for this metric [24]. Originally Dwivedi et al. [24] recommended constraining the models to have either 100K or 500k parameters for this benchmark (not all baselines state that they follow this). Since our model is less parameter efficient, compared to a standard GNN, due to the use of multiple MLPs for every step and concatenation

of various embeddings as input to those MLPs, we chose to use the 500k parameter budget. This corresponds to an embedding size of 88. In Table 7 we can see that this model performs comparably to GIN. If we instead would use an embedding size of 128, which would correspond to around 100k parameters for a traditional GNN architecture, such as GIN, AgentNet outperforms quite a few of the expressive baselines [51; 4; 71; 28; 8; 11; 9; 67] and half of the different ESAN [8] configurations which use different sub-graph construction techniques. Some of the baselines depend on explicitly pre-computed structural features, such as cycles [4; 28; 11; 9] which are known to be important in this task [24]. It is also worth noting that for some models such as GIN the performance on this task becomes worse when the parameter count is increased [24].

| Model | ZINC (MAE) |
|---|---|
| PPGN [51] | 0.256 ±0.054 |
| GIN [75] | 0.252 ±0.017 |
| PNA [18] | 0.188 ±0.004 |
| DGN [4] | 0.168 ±0.003 |
| HIMP [28] | 0.151 ±0.006 |
| SMP [71] | 0.138 ±? |
| ESAN (Edge Dropout) [8] | 0.172 ±0.005 |
| ESAN (Node Dropout) [8] | 0.166 ±0.004 |
| ESAN (Ego Graphs) [8] | 0.107 ±0.005 |
| ESAN (Ego Graphs + Root ID) [8] | 0.102 ±0.003 |
| GSN [11] | 0.108 ±0.018 |
| CIN-small [9] | 0.094 ±0.004 |
| CIN [9] | 0.079 ±0.006 |
| CRAWL [67] | 0.085 ±0.004 |
| AGENTNET | 0.258 ±0.033 |
| AGENTNET (Embedding size = 128) | 0.144 ±0.016 |

Table 7: Mean absolute error on the ZINC 12k dataset.

For the OGB experiments, we used the best hyperparameters from OGB-MolHIV (batch size of 64, 128 hidden units) and $k = 26$ agents with $\ell = 16$ steps for OGB-MolPCBA, while $k = 150$ agents with $\ell = 8$ steps were used for OGB-PPA, to account for larger graph size and memory constraints. As standard, results are reported over 10 random seeds [38]. For ZINC we used a batch size of 128, $\ell = 16$ steps, $k = 37$ agents, and either 90 or 128 hidden units and train it for 2000 epochs. In this case, we use 4 random seeds to report the results, following Dwivedi et al. [24].

## I IMPORTANCE OF DIFFERENT MODEL STEPS

In this section, we experimentally verify, that all of the model steps as described in Section 3 and Figure 1 are necessary. Besides testing Random Walk AgentNet, in Table 8 we also check how removing the node update step (AgentNet, No Node Update) or neighborhood update step (AgentNet, No Neighborhood Update) affects the model's ability to solve hard synthetic tasks. An important thing to note here is that normally, only nodes that have an agent on them perform the neighborhood update step. Meaning, that even if we remove the node update, there is still information accessible to the model about which neighboring nodes and how many times were visited, but its unknown by which agent. To account for this we also provide a model (AgentNet, No Node Update, and Neighborhood Update For All), which skips the node update step and performs neighborhood update for all nodes, not just ones that have an agent on them. As you can see in Table 8 this model is no longer expressive, while the other two versions do still have much better than random accuracy. However, we see that their predictions are no longer perfect. When we do not use the neighborhood update it becomes harder for the model to observe the structural information, especially if we want to find cliques, for which we need to check the neighborhood (Figure 1 (d)). Interestingly, AgentNet with no node update does quite well on all tasks, as visit counts can act as a surrogate for random features. Although performance is slightly worse than that of GIN with random features on all but the 2-WL task. In Table 9 we can see that these performance differences also transfer to the real-world OGB-MolHIV graph classification task. It is worth noting that random walk AgentNet achieves quite a close result to the full AgentNet. The reasons for this can be twofold: 1) as we have seen in Table 1 this formulation

| Model | 4-CYCLES [59] | CIRCULAR SKIP LINKS [15] | 2-WL |
|---|---|---|---|
| GIN [75] | 50.0 ±0.0 | 10.0 ±0.0 | 50.0 ±0.0 |
| GIN with random features [64; 1] | 99.7 ±0.4 | 95.8 ±2.1 | 92.4 ±1.6 |
| RANDOM WALK AGENTNET | **100.0** ±**0.0** | **100.0** ±**0.0** | 50.5 ±4.5 |
| SIMPLIFIED AGENTNET | **100.0** ±**0.0** | **100.0** ±**0.0** | **100.0** ±**0.0** |
| AGENTNET, No Neighborhood Update | 75.8 ±17.9 | 40.6 ±5.8 | 51.2 ±2.0 |
| AGENTNET, No Node Update | 92.4 ±9.7 | 89.8 ±8.9 | 98.9 ±0.6 |
| AGENTNET, No Node Update and Neighborhood Update For All | 50.0 ±0.0 | 10.0 ±0.8 | 49.7 ±1.2 |
| AGENTNET | **100.0** ±**0.0** | **100.0** ±**0.0** | **100.0** ±**0.0** |

Table 8: Ablation on how different AgentNet modules influence expressiveness on datasets unsolvable by 1-WL (GIN). Removing any of the AgentNet steps as described in Section 3 reduces models expressiveness.

| | OGB-MolHIV | |
|---|---|---|
| Model | Validation | Test |
| GIN [75] | 82.32 ±0.90 | 75.58 ±1.40 |
| GIN + virtual node [75] | 84.79 ±0.68 | 77.07 ±1.49 |
| ESAN [8]* | 84.28 ±0.90 | 78.00 ±1.42 |
| RANDOM WALK AGENTNET | 85.61 ±1.29 | 77.89 ±1.29 |
| AGENTNET, No Neighborhood Update | 84.06 ±0.76 | 75.41 ±1.41 |
| AGENTNET, No Node Update | 84.70 ±1.40 | 76.71 ±1.37 |
| AGENTNET, No Node Update and Neighborhood Update For All | 84.40 ±0.89 | 75.47 ±0.85 |
| AGENTNET | 84.77 ±0.92 | **78.33** ±**0.69** |

Table 9: Ablation on how different AgentNet modules influence test and validation ROC-AUC (%) on the OGB-MolHIV dataset. *Best result achieved by any version.

is able to recognize cycles, which are one of the most important structures in molecules; 2) as we have $k \approx n$ agents the whole graph can be explored even with random walks.

For synthetic tasks in Table 8 we used the same experimental setup as before (Appendix D.1) with a grid search over the hyperparameters, while for OGB-MolHIV in Table 9, we used the best hyperparameters of the full AgentNet (Appendix D.2).

## J    SELECTING THE NUMBER OF AGENTS

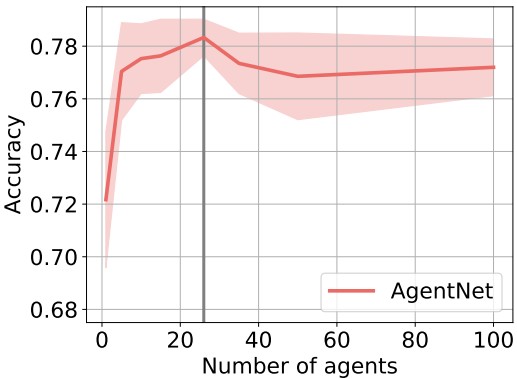

Figure 5: AgentNet ROC-AUC (%) on the OGB-MolHIV when using different number of agents (mean $n = 26$). Gray line marks the mean number of nodes in the dataset.

One of the hyperparameters that could be tuned for the model is the number of agents. To avoid an extensive grid search we normally set the number of agents to the mean number of nodes in the dataset. Besides matching the computational complexity of traditional GNNs, one reason for this is that we want that most of the nodes in the graph would be visited by at least some agent. Another reason is that node embedding vectors, where agents record information are of finite size. If we have

many more agents than nodes (e.g. a 1000 agents on a 10 node graph) there could be a negative interplay between the agents, where they would corrupt the information stored on nodes by other agents. Choosing to use around $n$ agents helps to avoid such negative interplay. As could be seen from Figure 2 (d), AgentNet performance is stable over a wide range of step and agent counts, but ideally, the model should visit the majority of the nodes if one wants to maximize accuracy, while further step and agent count increase beyond this does not bring much benefit. Here in Figure 5 we also provide an ablation on the effects the number of agents has on AgentNet ROC-AUC on OGB-MolHIV dataset, when using the best hyperparameters from the original experiment (Appendix D.2). We again see that having around $n$ agents is a good choice and that in fact, having $2n$ or $4n$ agents results in a slightly lower performance than when using less than $\frac{n}{2}$ agents. This again suggests that having only a fraction of the nodes active at a given time step can be a simple way to reduce the computational burden.

## K  IMPORTANT SUBGRAPHS

As we have discussed in the main paper, one of the potential benefits of AgentNet is that the agents can learn to prioritize important substructures. Here we investigate the MUTAG dataset [54], which is comprised of mutagenic and non-mutagenic molecules. It is known that a good indicator that a molecule is mutagenic is the presence of $NO_2$ groups [21]. This dataset was constructed such that all molecules have at least one $NO_2$ group, however, mutagenic molecules still tend to have more such groups. In Figure 6 we can indeed see that agents in AgentNet indeed visit such informative subgroups more often.

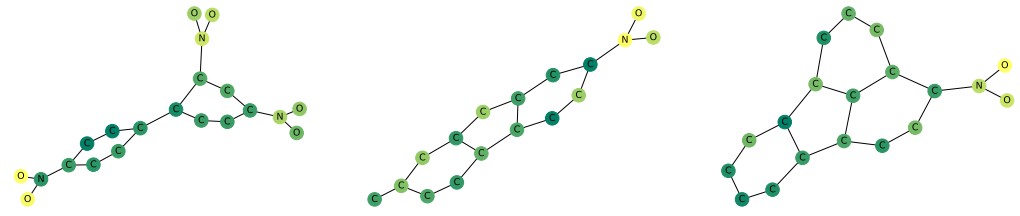

Figure 6: AgentNet node visitation heat map on test graphs from the MUTAG dataset. The brighter the color, the more often the given node has been visited. Agents prioritize some important substructures ($NO_2$) when they move around the graph.

