# OpenReview forum: "Agent-based Graph Neural Networks"
_ICLR.cc/2023/Conference — ICLR 2023 poster_

### Official Review · Reviewer_PGLs · 2022-10-23

**Confidence:** 3
**Correctness:** 4
**Technical Novelty And Significance:** 3
**Empirical Novelty And Significance:** 3
**Recommendation:** 8

**Clarity, Quality, Novelty And Reproducibility:**

- Clarity: the paper is clear and well written, the exposition is easy to follow and the organization of the content is good.
- Quality: the work is interesting and the analysis is complete. The results are not impressive but they are sufficient in light of the massive savings in computational cost.
- Novelty: the paper draws from well-known ideas in machine learning literature but the combination of these ideas is novel and well-motivated.
- Reproducibility: it should be possible to reproduce the results from the description of the method given in the paper and appendices. Code is provided as part of the supplementary material.

**Strength And Weaknesses:**

**Strengths**:
- The paper presents an interesting idea and is well-written and thorough.
- The theoretical analysis of the expressivity of AgentNets is interesting and easy to follow.
- The proposed method is significantly less expensive than other higher-order baselines and, although the performance is not always exceptional, I believe that it's more than acceptable.

**Weaknesses**:
- What is the difference between "pooling" and "readout" in the following sentence?
  > pooling is applied on the agent embeddings, followed by a readout function
- I suggest removing the claim that the agents "consciously select the next neighbor".
- The following claim:
  > In Table 3 we can see that AgentNet performs well in this scenario and outperforms even the best ESAN model.

  should be removed. There is no evidence to claim that the results reported in the table are significantly different between GIN (with virtual node), ESAN, and AgentNet.

**Summary Of The Paper:**

This paper presents a model for graph representation learning called AgentNet.

The main idea of the paper is to combine graph walks with neural networks: the model consists of agents that, in parallel, explore the graph by performing the following steps:

1. Updating the state of the current node
2. Aggregate information from the node's neighborhood
3. Update the state of the agent
4. Move to a new node in the neighborhood

The key difference from typical GNNs is that agents maintain their own global state as they explore the graphs, while also being able to "read" the other agents' contributions to the node states.

Through a nice theoretical analysis, the authors show that their model is more expressive than message-passing and even higher-order GNNs, while also having a cost sublinear in the number of nodes.

Experiments confirm that that model is indeed capable of solving synthetic benchmarks to measure the expressivity of GNNs at a fraction of the cost.
Results on real-world datasets are less exciting, although AgentNet shows performance comparable to much more expensive higher-order GNNs.

**Summary Of The Review:**

I really enjoyed the paper and I can safely recommend acceptance. There are a few minor issues that I ask the authors to address, but this is otherwise a good paper that would fit well at ICLR and would be of interest to the GNN community.

---

> ### Author Response · Authors · 2022-11-19
> **Response to reviewer PGLs**
>
> Thank you for your review! We have addressed the issues you outlined.
>
> >What is the difference between "pooling" and "readout" in the following sentence?
> pooling is applied on the agent embeddings, followed by a readout function
>
> Here pooling refers to pooling the agent embeddings (taking a mean of them) while readout refers to the MLP (function) that is applied on this averaged embedding. We’ll try to make this a bit more clear in the paper.
>
> >I suggest removing the claim that the agents "consciously select the next neighbor
>
> Thank you for the suggestion, we have rephrased the corresponding sentence.
>
> >The following claim:
> In Table 3 we can see that AgentNet performs well in this scenario and outperforms even the best ESAN model.
> should be removed. There is no evidence to claim that the results reported in the table are significantly different between GIN (with virtual node), ESAN, and AgentNet.”
>
> We agree that the difference isn’t very large so we changed the sentence.

---

> > ### Comment · Reviewer_PGLs · 2022-12-12
> > **Reply**
> >
> > I thank the authors for their reply. I have no further comments

---

### Official Review · Reviewer_z2nH · 2022-10-23

**Confidence:** 4
**Correctness:** 4
**Technical Novelty And Significance:** 4
**Empirical Novelty And Significance:** 2
**Recommendation:** 6

**Clarity, Quality, Novelty And Reproducibility:**

Paper is well presented, e.g. intuition about theoretical results is always given, which facilitates its understanding. Moving "away" from message-passing schemes is an extremely welcome novelty. Regarding reproducibility, the authors could present in the appendix a full description of hyperparameters and details about their architecture.

**Strength And Weaknesses:**

---Strength---
a) Novelty: moving away from message-passing schemes is a necessity of the graph learning community.
b) Clarity: presenting intuition together with a theoretical result is always welcome.
c) The authors precisely characterize the class of functions (graph properties) AgentNet can learn. This type of characterization is very help to guide practitioners' model choices. Eg., "deterministic functions of the r-hop neighborhood around a node v".
d) Results as far as I was able to follow and check all hold. Proofs are clear and well written.

---Weaknesses---
a) The main weakness in my opinion is the empirical evaluation. Since the authors tested their model in the OGB hiv task, I don't understand why other OGB datasets were not used. I don't find the TUDataset suit insightful of a practical evaluation of a model anymore. Further, the authors could have designed synthetic datasets that explore the properties known to be "learnable" by their model (see theory results). This could strengthen a lot the submission.

**Summary Of The Paper:**

This work proposes to learn graph-level representations by exploring the graph structure with an agent ---rather than with message-passing schemes. The authors present an extensive theoretical characterization of the expressive power of such a model, connecting it with, for instance, deterministic functions of the r-hop neighborhood around a node v. In practice, the authors show how the model is competitive with other baselines and confirm its enhanced expressive power.

**Summary Of The Review:**

Overall, the strength of this paper outweighs its weakness (see S&W section) and therefore my recommendation is supported. If the authors are able to at least partially address my concerns in the weakness section I'm willing to raise my score. To clarify, I'm not looking for SOTA results necessarily, but to be convinced that this architecture is competitive with existing ones.

---

> ### Author Response · Authors · 2022-11-19
> **Response to reviewer z2nH**
>
> Thank you for your review and valuable the suggestion. We think this helped us further solidify the paper.
>
> We originally only used OGB-MolHIV as that was the only dataset ESAN was tested on. We have now included the OGB-MolPCBA and OGB-PPA datasets as well as the ZINC graph regression dataset (Appendix H). We opted to skip the OGB-Code dataset, as it's a bit of a peculiar dataset, where all graphs are trees (normal MPNN 1-WL GNNs can perfectly capture tree graphs) and the task is actually a sequence prediction, not a classification task. Also note that we had experiments on QM9 in Appendix F.
>
> We also provide some more ablations in Appendices I and J that hopefully shed more light on how the model performs and how different parts of the model influence the results.
>
> As for the synthetic tasks, we feel that Table 1 captures well the results about learning to distinguish particular subgraphs, while Figure 2 a) showcases the sublinearity, which were the key theoretical claims.

---

> > ### Comment · Reviewer_z2nH · 2022-11-20
> > **Ack'ed rebuttal.**
> >
> > Thank you for acknowledging the suggestions and working on the submission. I will maintain my score but I definitely appreciate the effort, which has positively impacted my impression of the paper.

---

### Official Review · Reviewer_pbyu · 2022-10-26

**Confidence:** 5
**Correctness:** 2
**Technical Novelty And Significance:** 3
**Empirical Novelty And Significance:** 2
**Recommendation:** 5

**Clarity, Quality, Novelty And Reproducibility:**

The presented method is original, although it has some connection to other random-walk based methods.
The effectiveness of proposed method is limited of the randomness that hasn't been properly addressed.

**Strength And Weaknesses:**

**Strength**:
1. The angle of conducting agent-based random walk is very interesting. And with small number of agent, the method can be more efficient than message passing gnns.


**Weakness**:
1. Similar to Random Node Initialization based method, it achieves higher expressivity with the cost of introducing randomness which hurts training stability and generalization. The author doesn't give enough analysis and empirical result to show the badness of randomness introduced.
2. The theorems in section 4.1 is kind of questionable, giving it doesn't consider the badness of random. Specifically, for two isomorphism graphs, the traverse sequence for a agent can be different which leads to different embeddings that is not desirable. In the proof of theorem 2, the author claims that this kind of problem can be avoid by assuming there is a function that can assigning every possible IDDFS traversal to its isomorphism class of r-hop neighbor subgraph. The claim is misguiding as this is equivalent to say "Deep network can learn random noise" which is of course possible but needs tons of data and time to train and are not guranteed to converge easily. This is actually also observed in RNI based method.
3. All other theorems in section 4.1 have similar issues. The author starts every theorem of "there exists any configuration" which is perhaps true but also not considering the randomness of instability, hardness of training and requirements of more data.
4. In fact if isolating all randomness of random walks, the expressivity is known to be limited by 2-WL. Please see [Geerts 20]. Hence I highly doubt the expressivity of this designed agent-based graph neural network when taking randomness into account. Another related work that needs to compare is [Toenshoff et al. 21], which designing a very interesting subgraph encoding method with random walk.
5. The real-world performance is actually not good, which perhaps means that the expressivity of introduced method is limited, or suffering from the randomness. This means the proposed method is not practical to use for real-world problem.
6. The author also needs to include other datasets like ZINC and some larger dataset from Benchmarking GNN, as these datasets are observed to be correlated with expressivity.
7. The author claims that the agent can learn intelligent random walk, it would be very interesting to see some visualization of the walk learned, and to see whether it correlates to tasks like finding specific substructure.

[Geerts 20] Walk message passing neural networks and second-order graph neural networks
[Toenshoff et al. 21] Graph learning with 1d convolutions on random walks

**Summary Of The Paper:**

The paper proposes a new type of graph neural network with agents running learnable(intelligent) random walk on the given graph. The author studies the theoretical analysis of proposed agent based method in counting substructures, for both 1 agent case and multi agent case. To demonstrate the effectiveness, the author run experiments over counting substructure dataset and many small real-world TU dataset. The experimental results show improvement or comparable result to other GNNs.

**Summary Of The Review:**

To summarize, the paper studies applying learnable random walk on graphs for graph classification. Although being a different angle of performing representation learning, many issues of introduced randomness are not properly addressed, and the connection to other random-walk based graph neural networks mentioned above are not clear yet.


------------------------------------ after rebuttal -----------------
I would like to raise the score as 5, considering the great effort on providing more experiments and visualization. However, the real-world usage of the proposed method is still questionable. The expressivity of the proposed method is not well understood yet.

---

> ### Author Response · Authors · 2022-11-19
> **Response to reviewer pbyu (5/5)**
>
>
> >The real-world performance is actually not good, which perhaps means that the expressivity of introduced method is limited, or suffering from the randomness. This means the proposed method is not practical to use for real-world problem.
>
> It is true that our model performs comparably to other expressive baselines, but does not clearly outperform them. In this work, we provide more of a theoretical proposition of this new model that has certain nice properties. The experiments mainly serve to show that this model indeed works, can solve hard-to-solve tasks, and that under certain circumstances you can actually get away with visiting less than \\(n\\) nodes, even in practice. So in this light, performing as well as other expressive models is a satisfactory result. As we discussed in your previous question about randomness, it doesn’t seem that it would be causing too much trouble as witnessed by the fact that we achieve competitive results to other expressive models when using the same number of epochs.
> We also believe that performance of the approach could be further improved by several engineering tricks, but we felt such extensions would move the model further from theoretical analysis and the simpler model description we have presented.
>
> One potentially nice real-world use case for this would be when the graphs we are classifying are quite large, to the point that many other models start running into memory issues. We can already see this with REDDIT-BINARY and DD datasets, but in the wild, there are much larger graphs than the ones from those datasets (for example, there are many proteins and molecules with thousands of residues or atoms). Unfortunately, graph classification benchmark datasets with graphs much larger than REDDIT-BINARY or DD haven’t really been compiled yet.
>
> >The author also needs to include other datasets like ZINC and some larger dataset from Benchmarking GNN, as these datasets are observed to be correlated with expressivity.
>
> We have included ZINC and the other, much larger datasets from the OGB benchmark for graph classification.
>
> >The author claims that the agent can learn intelligent random walk, it would be very interesting to see some visualization of the walk learned, and to see whether it correlates to tasks like finding specific substructure.
>
> Thank you, this is a nice suggestion. We have added a visualization of the most visited nodes in the MUTAG dataset in Appendix K. It is known that NO2 groups tend to make molecules mutagenic and we can see in those heatmaps that agents tend to visit atoms of this group more often than other atoms in the molecule.

---

> > ### Comment · Reviewer_pbyu · 2022-11-24
> > **Thank you for detailed response**
> >
> > As a reviewer, I appreciate the author's detailed response, with additional experiments and visualization added. The proposed method is simple and I agree with the author that the method shows comparable performance with GIN on real-world dataset, with much better generalization ability than RNI method. Besides, I also agree with reviewer wMRf's comments. Overall, the designed idea is simple and interesting, despite the relative real-world performance is still not very good. Hence I would like to raise the score to 5.
> >
> > I encourage the author to explore combining this idea with CRAWL kind of idea, as CRAWL demonstrates significant performance improvement over real-world dataset. My main concern left is the real-world usage of the proposed method, which currently is limited by its real-world performance (I believe is affected by randomness, and the theoretical expressivity of the proposed method is not well studied yet). The author claim that the proposed method uses more intelligent random walk than CRAWL, I would expect to see the real-world performance improvement from the designed method.

---

> > > ### Author Response · Authors · 2022-11-30
> > > **Thank you for recognizing the improvements**
> > >
> > > Thank you for appreciating the improvements we have made to the paper and your suggestions that led to them.
> > >
> > > In regards to CRAWL, please note that AgentNet actually performs better than CRAWL in the majority of the datasets we tested (6 out of 7 TU-Datasets and 2 out of 3 OGB datasets).
> > >
> > > It is true, however, that AgentNet does not do that well on OGB-MolPCBA and ZINC datasets. As we discuss, a likely explanation for this would be that these two tasks focus on the global structure of the graph, which goes counter to the local, computationally efficient feature extraction we pursue with our approach (we discuss this task alignment in Appendix F). Performance, in this case, could potentially be improved by using more intricate global agent communication, for example, based on self-attention as suggested by reviewer wMRf. But we leave these various possible engineering optimizations for future work, as in this paper we aim to focus on theoretical analysis of the proposed method (Section 4) and showing that the desired features (sub-linearity, hard structure detection) are observed in practice (e.g. by synthetic tasks and real-world performance with few agents). Our results suggest that the current version of our method could already prove useful in several practical applications, particularly for the more efficient classification of large graphs (with only a few agents), when existing approaches are too memory intensive.
> > >
> > > (edited for better readability)

---

> ### Author Response · Authors · 2022-11-19
> **Response to reviewer pbyu (4/5)**
>
> >Another related work that needs to compare is [Toenshoff et al. 21], which designing a very interesting subgraph encoding method with random walk.
>
> This is indeed a very interesting work with some similarities to our approach. In particular, the CRAWL architecture of [Toenshoff et al. 21] explicitly adds node identification and adjacency information to the model; this allows for a recognition of specific nodes/edges, and results in very strong expressiveness, similar to our case.
>
> The main advantage of AgentNet to the approach of CRAWL [Toenshoff et al. 21] is that CRAWL is based on sampling random walks from the graph, whereas AgentNet can learn to follow more sophisticated paths when navigating through the neighborhood. For instance, consider an application where the key substructure to recognize is a simple path of length \\(L\\), such that all nodes of this path are marked by a special node feature \\(a_0\\). If the copies of this structure are embedded in a dense graph, then CRAWL will have a very low probability of traversing such a path from beginning to end, since the next neighbor visit is always selected uniformly at random. In contrast to this, AgentNet can learn that the nodes labeled with feature \\(a_0\\) are crucial for this task, and hence if an agent encounters such a node, it can proceed to only visit its neighbors that are also marked with \\(a_0\\), hence efficiently traversing the path and identifying this structure. Also, CRAWL as presented, updates all \\(n\\) nodes in every layer and pools all of the \\(n\\) nodes at the end for graph classification, which makes its complexity depends on the number of nodes.
>
> We have included this model in the result comparison.

---

> ### Author Response · Authors · 2022-11-19
> **Response to reviewer pbyu (3/5)**
>
> >In fact if isolating all randomness of random walks, the expressivity is known to be limited by 2-WL. Please see [Geerts 20]. Hence I highly doubt the expressivity of this designed agent-based graph neural network when taking randomness into account.
>
> The reason why the expressive power of walkMPNNs [Geerts 20] is limited by 2-WL is that in each in round, the embedding of a node pair \\( (u,v) \\) in walkMPNN is updated based on the multiset of *all* possible walks of length \\( \ell \\) from \\(u\\) to \\(v\\). Due to the properties of 2-WL, when two node pairs are equivalent under 2-WL, then this multiset of walks between \\(u\\) and \\(v\\) is also identical.
> In contrast to this, our approach allows the agent to apply more sophisticated strategies, by (1) learning to ‘intelligently’ (non-uniformly) select the next neighbor to visit in each step, and by (2) ‘marking’ the observed nodes along the walk, and hence later recognizing a node that has already been visited before during the walk. Together, this provides significantly larger expressive power.
>
> For a concrete example, consider the Rook’s 4x4 and Shrikhande graphs, which are the most well-known example for indistinguishability by 2-WL. In both of these graphs, between any pair of nodes, there are exactly 2 distinct walks of length 2. As such, if all node pairs begin with the same features, then the embeddings in a walkMPNN are updated based on the same multiset of two walks of length 2, for all of the node pairs in both graphs. As such, all node pairs will have the same embedding after any number of rounds, and thus the graphs cannot be distinguished.
>
> In contrast to this, an agent in our approach can distinguish the two graphs, as we have discussed in our Theorem 2 and Corollary 3. In particular, the agent can learn an exploration strategy where it visits all 6 neighbors of the starting node \\(v\\) in only 11 steps, in either of the two graphs. If the agent leaves a distinct marking at each node, then it can deduce which of the visited nodes are adjacent to each other: if two neighbors \\(u_1\\), \\(u_2\\) of \\(v\\) are adjacent, and \\(u_1\\) is visited first, then the agent can observe the distinct marking of \\(u_1\\) in the message passing step at \\(u_2\\).
>
> In a Rook’s 4x4 graph, each node is contained in two 4-cliques, while in the Shrikhande graph, there is no 4-clique at all. When the last node of such a 4-clique is visited, the agent can recognize this situation in Rook’s 4x4: it will observe that there are 3 distinct neighbors \\(u_1\\), \\(u_2\\), \\(u_3\\) of \\(v\\) that are also pairwise adjacent to each other. On the other hand, this will never happen in the Shrikhande graph. As such, the agent can learn to execute a walk of length 11 that produces a different final embedding in the two graphs.
>
> Another way to look at this is that as the agent walks the graph, it can assign unique IDs to each node it visits (e.g. visited by Agent 1 at time-step 2). As known from [Loukas2020] MPNN with unique node IDs is universal. When an agent walks a subgraph, it can exactly capture the structure between all nodes that it visited, due to them being uniquely identifiable when neighborhood aggregation is performed.
>
> [Loukas2020] Andreas Loukas. What graph neural networks cannot learn: depth vs width. ICLR (2020)

---

> ### Author Response · Authors · 2022-11-19
> **Response to reviewer pbyu (2/5)**
>
> >The theorems in section 4.1 is kind of questionable, giving it doesn't consider the badness of random. Specifically, for two isomorphism graphs, the traverse sequence for a agent can be different which leads to different embeddings that is not desirable. In the proof of theorem 2, the author claims that this kind of problem can be avoid by assuming there is a function that can assigning every possible IDDFS traversal to its isomorphism class of r-hop neighbor subgraph. The claim is misguiding as this is equivalent to say "Deep network can learn random noise" which is of course possible but needs tons of data and time to train and are not guaranteed to converge easily. This is actually also observed in RNI based method.”
>
> AND
>
> >All other theorems in section 4.1 have similar issues. The author starts every theorem of "there exists any configuration" which is perhaps true but also not considering the randomness of instability, hardness of training and requirements of more data.”
>
> We agree that this is indeed a relevant question.
>
> The theoretical study of GNN expressiveness is a relatively young area. For most of the GNN variants developed, the authors have focused on the maximal expressiveness of the new model, i.e. which graphs can or cannot be distinguished by the model at all (as in e.g. the 1-WL upper bound for regular GNNs) since this is the first natural question when studying a new GNN variant. In order to understand this, we need to analyze implementations that push the given model to “its limits”, even though these implementations might be unlikely to actually appear in practice.
>
> In this regard, our work takes an identical approach to the most important works in the area, such as the works of Xu et al. [Xu19], Maron et al. [Maron19], or Sato et al. [Sato19]. These works also all apply similarly “impractical” constructions to study the maximal expressiveness of specific GNN models; in particular, their theoretical analysis specifically invokes the universal approximation theorem, just like our work.
>
> Of course, once the maximal expressiveness of a given GNN model is understood, it is indeed an important follow-up topic to study whether GNNs are actually able to learn specific kinds of functions in practice and to devise theoretical bounds on the aggregation and update functions required in the GNNs to do so. We agree that this is a crucial direction for future work on GNNs in general. Until such techniques are available, most works on GNNs (including ours) use experiments on challenging synthetic datasets to demonstrate that a given GNN model can indeed learn functions that raise its expressive power well beyond 1-WL. Table 1 in our work demonstrates that AgentNet can indeed successfully solve such tasks.
>
> Hence, we believe that our approach for the theoretical and empirical analysis of the model is in line with the other defining works in the area. A deeper theoretical analysis of whether concrete functions can indeed be learned within our model is an interesting and important question but might require further theoretical tools that are still in development for this area.
>
> [Xu19] K. Xu, W. Hu, J. Leskovec, and S. Jegelka. How powerful are graph neural networks? ICLR 2019.
>
> [Maron19] Maron, H., Ben-Hamu, H., Serviansky, H., and Lipman, Y. Provably powerful graph networks. NeurIPS 2019.
>
> [Sato19] R. Sato, M. Yamada, and H. Kashima. Approximation Ratios of Graph Neural Networks for Combinatorial Problems. NeurIPS 2019.

---

> ### Author Response · Authors · 2022-11-19
> **Response to reviewer pbyu (1/5)**
>
> We thank you for your review and for pointing us to the papers [Geerts 20] and [Toenshoff et al. 21] – we have added them both as references to our paper. We respond to your concerns below and in the next few comments.
>
> >Similar to Random Node Initialization based method, it achieves higher expressivity with the cost of introducing randomness which hurts training stability and generalization. The author doesn't give enough analysis and empirical result to show the badness of randomness introduced.
>
> While some of the decisions of the agent are indeed randomized, we do not believe that randomness plays such a defining role in our model. In particular, after the agent is well-trained, the decisions in the transition step are mostly based on the transition function learned (and hence on the current embedding of the agent and its neighbors). In the extreme case, if the agent learns the IDDFS traversal described in Lemma 1, then the general exploration strategy of the agent will not depend on randomness anymore; randomness will only play a role in selecting the order to explore the unvisited neighbors of each node. In that sense, our approach is very different from e.g. a random walk-based approach.
>
> It is true, however, that random agent placement might make training of the model a bit harder and slower than for traditional GNNs. However, as we see from Table 1 the AgentNet happens to converge to always distinguishing the relevant graphs, while GIN with random features struggles. To more clearly showcase this, we include the test accuracy curves as a function of training steps in Appendix G. There you can see that AgentNet always converges faster than GIN with random features and as tasks get harder (e.g. 2-WL task) the difference increases even further.
>
> Also, note that for the TU-Datasets (Table 2)  and OGB (Table 3 and Table 6) we used the standard number of training epochs (350 for TU-Datasets and 100 for OGB datasets) when training our models. So the empirical results are achieved with the same training budget as had originally been used for the baselines. The results compare well to the baselines, so there is no reason to think that randomness is too detrimental.

---

### Official Review · Reviewer_6DSe · 2022-11-03

**Confidence:** 3
**Correctness:** 3
**Technical Novelty And Significance:** 3
**Empirical Novelty And Significance:** 3
**Recommendation:** 6

**Clarity, Quality, Novelty And Reproducibility:**

The paper is easy to follow with clear motivation.

The proposed method is not technically nontrivial, but is well justified with good properties and is flexible to extend. The idea is novel among the studies based on the message-passing mechanisms.

The codes are provided for reproducibility.


**Strength And Weaknesses:**

**Strength**
- This paper provides a novel perspective for graph based learning beyond message-passing mechanism
- The proposed model enjoys several good properties, e.g. expressiveness and efficiency, which are demonstrated via theoretical and empirical evidence
- The new architecture is practical and flexible to extend

**Weaknesses**
- Lack ablation study to verify the importance of different steps in the model

**Questions**
- Do the agents share the same set of model parameters, but only vary in the start node? I am curious if these agents can be designed to collect different patterns.
- Will the proposed method also encounter the oversmoothing or oversquashing issue if the number of steps or the number of agents are set too large, and undereaching issue if they are set too small?
- Can the proposed method be extended to serve for other popular downstream tasks, such as node classification and link prediction tasks?
- It would be better if ablation study can be provided to show the importance of each model design, e.g. node update, neighborhood aggregation, etc.


**Summary Of The Paper:**

This paper introduces an agent-based GNN which differs fundamentally from the message-passing based architecture. AgentNet consists of a collection of neural agents that are trained to walk on the graph and their states are pooled to make graph classification predictions. Theoretical analysis and empirically evaluation verify good properties of the proposed model.


**Summary Of The Review:**

This paper is overall well motivated and well written. The proposed agent based graph learning architecture is novel, fresh, flexible, efficient and expressive. It should be interesting to the community.

---

> ### Author Response · Authors · 2022-11-19
> **Answer to reviewer 6DSe**
>
> Thank you for your insightful questions and suggestions.
> We have now included the ablation of different steps of the proposed AgentNet in Appendix I. There we also discuss why these parts are necessary. Essentially, the expressive power of the model would suffer if we drop any of them, as was also suggested by for example the discussed examples of how cliques or cycles could be detected by AgentNet.
>
> >Do the agents share the same set of model parameters, but only vary in the start node? I am curious if these agents can be designed to collect different patterns.
>
> All the MLPs used by the agents share the parameters. But each agent has a learnable initial embedding (its memory), similar to a dictionary entry in a large language model. So technically different agents could learn to do somewhat different things, based on their initial embedding. However, since the agents are placed randomly on the graph there isn’t that much pressure encouraging them to learn different things as they need to be agnostic to their placement.
>
> >Will the proposed method also encounter the oversmoothing or oversquashing issue if the number of steps or the number of agents are set too large, and undereaching issue if they are set too small?
>
> Of course, the agents can still suffer from underreaching. If the agent needs to know the distance between node A and node B to make the correct prediction if it does not do enough steps this would be problematic. Even if we would have say one agent starting from A and another from B there is a distance where they could never meet.
>
> As for oversmoothing and oversquashing, the fact that each agent carries memory with itself somewhat lessens the negative impact of having many agents. However, one can imagine that if we have 100 times more agents than nodes and if the ability for agents to record information on each node is very important for solving the task, problems could arise due to a clash of information, unless very wide embeddings are used. Since we don’t perform sufficiently detailed experiments on oversmoothing and oversquashing, we have decided to remove the discussion of them from the paper. Also, nowadays we have various remedies for these problems for normal GNNs such as various normalization techniques and improved convolutions so they might be a bit less relevant.
> Because the architecture is recurrent over the timesteps, it is also possible that some gradient and value instability issues seen with RNNs when thousands of steps are used could also arise, even though we haven’t observed them in our experiments. Of course, the common remedies used in RNNs, such as making use of LSTM cells, could help in such a case. But in the experiments, we see that even with tens of steps it's possible to achieve good results.
>
>
> >Can the proposed method be extended to serve for other popular downstream tasks, such as node classification and link prediction tasks?
>
> Such an extension could certainly be possible, and it would be an interesting direction for future work to figure out how to best do this. Trivially, instead of using the agent embeddings to make the final graph-level prediction, one could instead use the node embeddings to classify the nodes or compute a similarity between those node embeddings to predict links. Of course in this case we would have to drop the model sub-linearity (w.r.t. number of nodes) as we would need each node to be visited by some agent, ideally many times. But such a model would have some similarity with diffusion on graphs [Gasteiger2019], which does work quite well for node classification and link prediction. Another potentially interesting extension would be looking at some graph optimization problems, such as TSP. Ant colony optimization algorithms have seen some success there, and those ‘ants’ are a bit like simplified versions of our agents.
>
> [Gasteiger2019] Johannes Gasteiger, Stefan Weißenberger, Stephan Günnemann. Diffusion Improves Graph Learning. ICLR (2019)

---

### Official Review · Reviewer_wMRf · 2022-11-04

**Confidence:** 4
**Correctness:** 3
**Technical Novelty And Significance:** 2
**Empirical Novelty And Significance:** 2
**Recommendation:** 5

**Clarity, Quality, Novelty And Reproducibility:**

- Originality is limited in that most part of architectures still follow conventional message passing except the introduction of meta-level information carrier in the form of agent(s). Also the marginal gains in the experiments adds less value to the novelty of the proposed approach.
- While the overall paper is readable, there are few points that are not clear from the writing:
    - Why isn’t the proposed approach also translate into a BFS and only DFS?
    - Before theorem 2, what do authors mean when they mention “ AgentNet can identify all edges between any two visited nodes? Doe you mean a path or multiple edges between two nodes?
    - In multiple agent case, It is not clear why just having random vector initializations as agent id’s ensure that agents become expressive enough to disentangle their own information at the end of learning. Why is there not a possibility that all or many agents end up learning similar information. Eventually, one would expect that agent information will be dominated by information in the nodes it traverses.
    - It is not clear how to choose between single agent and multiple agent architectures for a given graph and problem. It appears that for most cases single agent design is not useful and that is only important for showing theoretical analysis.
    - In multiple-agent case, it seems that pooling may have big impact and ablation related to different pooling methods is useful for clarity.
- The quality of the paper is fair. Many of the content in the main paper is not important (for example the propositions mentioned above), while the appendix contains a lot of proofs and architectural details. It would help to reorganize the content so as to highlight important contributions in the main paper. The technical quality of the paper can be further improved with the inclusion of the cited comparison and showing more concrete results on the theoretical and empirical gain of multiple agent scenario.


**Strength And Weaknesses:**

Strengths:
------------
+ The topic of designing expressive graph neural networks for learning on graphs has received significant attention recently and if of great interest to community.
+ The use of meta-level information in the form of agents to gain subgraph-level expressiveness is an interesting idea and has potential for more exploration.
+ The property of the proposed architecture to provide sublinear computational complexity is very appealing.
+ The paper shows strong empirical results in the settings considered by the authors  with gains either in performance or in complexity over previous baselines
+ The ablations with respect to the number of agents and number of steps are very insightful and clearly help to discern if and when the proposed method achieves
gain.

Weaknesses
---------------

- While the idea of intelligent walk is interesting, the language of agent is not useful. The single agent case is just a single initialization of an intelligent walk with memory and multi agent case is its extention to multiple walks. It is also confusing to present polling as some sort of global communication, it is just better to present it as aggregation of information from multiple walks.
- As the transition function is based on attention, this method is clearly very related to attention based graph neural networks. Of course, attention [1] is one way to select a subgraph to focus on for computing a node embedding and thereby graph emebdding. The authors fail to discuss how the proposed  method relates to attention based models. Both theoretical and empirical justification is required.
- There are several strong works that are relevant to this approach and provide both theoretical and empirical insights [2,3,4]. The authors have neither discussed them nor considered them for comparisons which is a big miss.
- While the theoretical results attempt to detail different scenarios, many of the theorems and lemmas are more of propositions or insights (e.g. Thm 2, Corollary 3, Lemma 4, thm 7, lemma 8) and it would be good to describe them so.
- While the approach can be considered independent of graph size in some cases (not really clear if you can always avoid visiting most nodes), it does seem to depend on the degree of nodes and hence large graphs graphs where several (even ~100) nodes may have high degree, this approach will potentially struggle. Can the authors comment on such scenarios
- More concrete theoretical results relating to the counting of substructures such as cycle and cliques are provided for single agent case. how does this results hold in multiple agent case?
- In experiments, using mean number of nodes as number of agents is a very strange and adhoc choice for the proposed approach. It needs better motivation than just matching the complexity of other  works.
- Interpretations of the results are not always clear or adequate - For figure 2(d) for example, higher number of agents of course lead to fewer node visits per agent and it does not perform better than GIN. So how to interpret these results? The claim of better robustness using Table 3 is very hand wavy and needs lot more investigation. Can the authors comment more on why they think those results indicate robustness to oversmoothing and oversquashing?
- Empirically, the gains with AgentNet are below marginal and couple this with missing comparisons, the proposed approach has lot of room for improvement.

[1] HOW TO FIND YOUR FRIENDLY NEIGHBORHOOD: GRAPH ATTENTION DESIGN WITH SELF-SUPERVISION, KIM et. al. ICLR 2021

[2] Random Walk Graph Neural Networks, Nikolentzos et. al. Neurips 2020

[3] Walk Message Passing Neural Networks and Second-Order Graph Neural Networks, Gertz et. al. 2020

[4] Ordered Subgraph Aggregation Networks, Qian et. al. June 2022

**Summary Of The Paper:**

This work focuses on the design of expressive neural network architecture for  learning on graphs, specifically aimed at improving performance on graph-level tasks. The proposed approach modifies the existing message passing mechanism by introducing the concept of an agent(s) capable of aggregating different types of graph (node, edge) information at subgraph level and then mapping it to graph level output. The key ability of such agent(s) is to perform intelligent walks over subgraphs of size l. The authors argue that such a construction enables sublinearity in learning algorithms that is independent of original graph size. The authors perform theoretical analysis of various properties of this new design in settings with both single and multiple agents. The main theoretical results revolve around the ability of the new approach to distinguish non-isomorphic subgraphs and  to count various substructures such as cycles and cliques. The authors also present and discuss tradeoff scenarios on using single vs multiple agents architectures. Three synthetic datasets are used to test these theoretical properties and compared with various representative baselines focussing on expressivity of graph neural networks. To fully elucidate the gains made in computational complexity and asses the learning performance of the proposed design, the authors present graph classification results on  a suite of real-world graphs and compare it against representative baselines.

**Summary Of The Review:**

The topic of designing expressive graph neural network is very interesting and the authors present an interesting idea towards that direction. The claims of sub-linearity is appealing and the authors show some improvement on classification tasks over representative baselines. However, both the theoretical results and empirical evidence fall short of strongly supporting overall claims of the paper and further the complete miss of discussion with attention based approaches and other related works, both of which informs my current assessment of the paper.

---

> ### Author Response · Authors · 2022-11-19
> **Response to reviewer wMRf (6/6)**
>
>
> >Why isn’t the proposed approach also translate into a BFS and only DFS?
>
> The agents perform a walk in the graph, i.e. in each step, they can only move to a neighbor of the current node. This does not allow a BFS-style traversal; if we list the nodes of a graph in a BFS order, then the consecutive nodes in this order will often not be adjacent to each other and can be at a rather large distance in the graph. Instead, we require an exploration strategy that systematically traverses the neighborhood of the starting node by moving to an adjacent node in each step. The Iteratively Deepening DFS is a natural choice in such a setting.
>
>
> >Before theorem 2, what do authors mean when they mention “ AgentNet can identify all edges between any two visited nodes? Do you mean a path or multiple edges between two nodes?
>
> We mean that the agent can recognize the existence of any edge upon visiting both of its endpoints; that is, if \\(u\\) and \\(v\\) are two nodes visited by the agent, then the agent can deduce whether \\(u\\) and \\(v\\) are adjacent to each other. This implies that the agent is expressive enough to identify the subgraph induced by all the nodes it has visited.
>
>
>
>
> >In multiple agent case, It is not clear why just having random vector initializations as agent id’s ensure that agents become expressive enough to disentangle their own information at the end of learning. Why is there not a possibility that all or many agents end up learning similar information. Eventually, one would expect that agent information will be dominated by information in the nodes it traverses.
>
> Indeed, in practice, we also expect that the agent's information is mostly dominated by the nodes it has traversed.
> Lemma 8 is more of a theoretical tool; it shows that the agents do have the expressive power to execute this disentanglement in theory. This provides a straightforward way to extend the expressiveness results of Section 4.1 to the multi-agent case, without the further technical difficulty of having to analyze how the possible interaction of agents affects each step of these proofs.
> Please note though, that in our implementation agent embeddings at the start of the training are initialized randomly but they are trainable. Similarly to dictionary embeddings in large language models. So if disentanglement is crucial for the task, AgentNet could learn slightly more informative initial embeddings for the agents (each one of the \\(k\\) agents has its own learnable randomly initialized embedding vector).
>
>
>
>
> >It is not clear how to choose between single agent and multiple agent architectures for a given graph and problem. It appears that for most cases single agent design is not useful and that is only important for showing theoretical analysis.
>
> Indeed, we agree that the single-agent case is more interesting from a theoretical perspective; it ensures that the agent only observes its own past markings in the graph, making the analysis of the model significantly less technical. For practical purposes, one would probably use the multi-agent setting instead in most cases.
>
>
> >In multiple-agent case, it seems that pooling may have big impact and ablation related to different pooling methods is useful for clarity.
>
> Yes, one way to potentially improve the model could be to investigate different pooling techniques. However, sum pooling is known to be expressive and usually works well for various GNN methods, and is used in the majority of cases. Thinking about this from an algorithmic perspective, the two main ways multiple agents can classify a graph is either by doing majority voting (sum/mean pooling) or by checking if at least one of them found a ‘proof’ of a particular class (max pooling). So to allow the model to deal with both scenarios in practice we use both pooling methods and concatenate the resulting outputs when making the graph-level prediction,
> even though using sum and max together only offered a small improvement over just using sum. As we didn’t see a theoretical motivation to try anything else, we simply stuck with this, but investigating other approaches could be an interesting extension.

---

> ### Author Response · Authors · 2022-11-19
> **Response to reviewer wMRf (5/6)**
>
> >Interpretations of the results are not always clear or adequate - For figure 2(d) for example, higher number of agents of course lead to fewer node visits per agent and it does not perform better than GIN. So how to interpret these results? The claim of better robustness using Table 3 is very hand wavy and needs lot more investigation. Can the authors comment more on why they think those results indicate robustness to oversmoothing and oversquashing?
>
> We are unsure how to understand your first example “For figure 2(d) for example, *higher number of agents* of course *lead to fewer node visits per agent* and it does not perform better than GIN.” In the grid in Figure 2 (d) the number of agents is adjusted independently from the number of steps. If we take one row, the number of steps is constant (each agent moves this number of steps) and the number of agents keeps increasing, so the total number of nodes visited is proportional to the number of agents times the number of steps (of course at some point nodes are visited repeatedly). GIN as standard on TU datasets used 4 layers here. So it’s comparable to performing \\( 4*n\\) node visits with AgentNet. All configurations that perform fewer node visits than this, are marked in gray (when at least  \\(n\\)  node visits are performed) and black (when less than  \\(n\\)  node visits are performed). We see that all gray cells have higher accuracy than GIN while performing fewer steps. Now if we go below  \\(n\\)  node visits (meaning we go towards the lower right corner) the accuracy indeed decreases, but we are visiting way fewer nodes. With just 8 agents and 8 steps each, we can match the GIN accuracy of 77%, while visiting only up to 64 nodes, while graphs have 284 nodes on average. Of course, if we use only 2 agents and 2 steps, we visit only 4 nodes, so the accuracy is quite low, but we can’t really hope to correctly classify the graph when only seeing 4 nodes out of 284.
> When we move towards the upper right corner (we are increasing the number of node visits), the accuracy settles around 79% as soon as we have a good chance of visiting the majority of the nodes. The argument for some robustness to oversmoothing and oversquashing was that the accuracy stays flat over a wide range of high agent and step counts, while standard GNNs would often see a noticeable decline in accuracy when a larger than the optimal number of layers was used, as originally shown by [Kipf2017]. However, you are right, that to make such claims we would need to perform more extensive experiments tailor-made for this. As this is far from the main point of the paper, we will just forgo the discussion of oversmoothing and oversquashing. Thank you a lot for pointing this out.
>
> [Kipf2017] Thomas N Kipf, Max Welling. Semi-supervised classification with graph convolutional networks. ICLR (2017)
>
>
> >Empirically, the gains with AgentNet are below marginal and couple this with missing comparisons, the proposed approach has lot of room for improvement.
>
> The main point of the experiments is to show that this proposed architecture, which can be fully independent of the number of nodes in the graph, indeed works and that certain nice theoretical features such as potential sublinearity and the ability to distinguish hard-to-distinguish graphs can be verified experimentally. For this, we choose to compare to expressive baselines (as our model also provides better expressive power) that usually have higher computational complexity. So we do not really hope to substantially outperform them and we think that performing as well as they do is already nice. The goal of the paper was not to introduce a state of the art architecture but to show that another type of graph-classification architecture can have some nice properties and works well. Note that the model does consistently perform better than GIN, which shares the computational complexity and neighborhood aggregation style (function). Even with just one agent, the performance is often comparable. The synthetic benchmarks also back our theoretical claims.

---

> ### Author Response · Authors · 2022-11-19
> **Response to reviewer wMRf (4/6)**
>
>
> >While the approach can be considered independent of graph size in some cases (not really clear if you can always avoid visiting most nodes), it does seem to depend on the degree of nodes and hence large graphs where several (even ~100) nodes may have high degree, this approach will potentially struggle. Can the authors comment on such scenarios
>
> Yes, the approach is indeed dependent on the node degree, due to neighborhood pooling and next-hop selection. But this is normally almost impossible to avoid in GNNs and generally distributed computing algorithms. If the graph as a whole is very dense (number of edges \\( ~ n^2\\)), then of course our approach will be much less efficient. But normally the graphs considered for GNNs are relatively sparse, even if there are some high-degree nodes. In fact, the REDDIT-BINARY dataset we tested on, has some very high degree nodes (the largest node degree is 3062, see Table 4 for graph statistics) and our model performs quite well on it. However, one slight inefficiency that can arise in this case is if many agents end up on the high degree node, each agent will perform attention calculation over this large neighborhood, which can enlarge the memory footprint compared to a model that performs only one neighborhood investigation for this one node. Although, if needed, this could also be addressed, by slightly simplifying the model (Appendix C.5) and computing one transition probability distribution for all agents that are on the same node.
>
> We see from Figure 2 and the performance of AgentNet with one agent in Table 2 that indeed, we can only avoid visiting most nodes to an extent; this will largely depend on the dataset and the task you are trying to solve. But still, it is nice to see that if needed we can reduce the complexity if for example the graph is very big or the memory is constrained.
>
> >More concrete theoretical results relating to the counting of substructures such as cycle and cliques are provided for single agent case. how does this results hold in multiple agent case?
>
> Note that Lemma 8 describes an agent implementation that does not interfere with the remaining agents. This technique can be combined with the theorems in Section 4.1. to generalize all these claims on recognizing substructures to the multi-agent case (with the caveat that if two agents start very close to each other, then a given substructure might be counted by both).
>
>
> >In experiments, using mean number of nodes as number of agents is a very strange and adhoc choice for the proposed approach. It needs better motivation than just matching the complexity of other works.
>
> It is true that the number of agents could be tuned for each task or the available computational budget. But that would introduce a lot of parameters one would grid-search over, which is a bit unfair to the other models. The other motivation, besides just matching the computational complexity of GIN, is that to achieve strong results, we ideally want that all nodes would be observed by at least some agent. Having \\(n\\) agents helps to ensure this. Also, while having more than \\(n\\) agents could potentially benefit the model performance, at least in some cases, one could imagine that for example, if we had 1000 agents on a 10-node graph, none of the agents could really successfully record information on nodes due to large ‘interference’ from other agents. So choosing to use \\(n\\) agents should also be a safe choice that should avoid too much negative interplay between the agents.
>
> We do now provide this discussion and a simple ablation on the number of agents for OGB-MolHIV in Appendix J.

---

> ### Author Response · Authors · 2022-11-19
> **Response to reviewer wMRf (3/6)**
>
>
>
> >There are several strong works that are relevant to this approach and provide both theoretical and empirical insights [2,3,4]. The authors have neither discussed them nor considered them for comparisons which is a big miss.
>
> Indeed, there is a very rapidly growing body of work related to GNN expressiveness – we are grateful for any such papers we might have missed. We added the above references to our paper, and discuss them in slightly more detail below.
>
> The works of [2] and [3] are both based on random walks in the graph. The work of [2] uses the well-studied random walk kernel to compare the input graph to a number of trainable ‘hidden graphs’, hence indirectly using the number of matching random walks. The result is then fed into a standard fully connected neural network. The work of [3], on the other hand, updates the embedding of node pairs based on the multiset of all random walks between the two nodes. This paper [3] was also mentioned by Reviewer pbyu; we also discuss it in detail in our answer to Reviewer pbyu.
>
> As the work of [3] already shows, such a random walk-based approach is upper bounded in expressiveness by 2-WL, since, roughly speaking, 2-WL-equivalent graphs have the same set of random walks. In contrast to this, our AgentNet model has significantly larger expressive power: it can also distinguish graphs that are equivalent under 2-WL, and in general, any two distinct neighborhoods (with a sufficiently large number of steps). Intuitively, this is because the marking of nodes allows the agent to recognize specific nodes, i.e. decide whether it is visiting the same node in two given steps; this is not possible in the random walk-based models. Furthermore, on the more practical side, another strong advantage of AgentNets is that they can learn to navigate the graph in a more sophisticated way than random walks, focusing e.g. on nodes with features that are critical for a given application. In our experiments, we also observe that AgentNet, with an intelligent exploration strategy, manages to distinguish the 2-WL indistinguishable graphs, while the random walk version fails to do so.
>
> In contrast to this, the work of [4] introduces a new subgraph-enhanced GNN framework that can be considered a generalization of several recent GNN variants. However, this is a rather different approach from our work; moreover, its time complexity scales with \\(n^k\\) by default, where \\(k\\) is the size of the subgraphs being considered.
>
> [2] Random Walk Graph Neural Networks, Nikolentzos et. al. Neurips 2020
>
> [3] Walk Message Passing Neural Networks and Second-Order Graph Neural Networks, Geerts et. al. 2020
>
> [4] Ordered Subgraph Aggregation Networks, Qian et. al. June 2022
>
> [Huang2021] Ningyuan Teres Huang and Soledad Villar. "A Short Tutorial on The Weisfeiler-Lehman Test And Its Variants." ICASSP 2021
>
>
>
>
> >While the theoretical results attempt to detail different scenarios, many of the theorems and lemmas are more of propositions or insights (e.g. Thm 2, Corollary 3, Lemma 4, thm 7, lemma 8) and it would be good to describe them so.
>
> Some of these claims are indeed on the simpler side and can be proven in only 1-2 paragraphs. However, in line with some other theoretical works, we believe that insights or propositions should mostly refer to even simpler statements, which can be justified in at most 1-2 sentences. Since the proof of these claims certainly requires some more explanation, we decided to refer to them as lemmas or theorems.

---

> ### Author Response · Authors · 2022-11-19
> **Response to reviewer wMRf (2/6)**
>
>
> >As the transition function is based on attention, this method is clearly very related to attention based graph neural networks. Of course, attention [1] is one way to select a subgraph to focus on for computing a node embedding and thereby graph embedding. The authors fail to discuss how the proposed method relates to attention based models. Both theoretical and empirical justification is required.
>
> You are right, there certainly is some relation to other attention-based models as we do also use attention and we will include this discussion in the paper.
> The main way the attention is used is as another type of graph convolution (e.g. GAT [Veličković2018]) as a learnable/better way to do mean pooling. Such models use the whole graph with only effectively soft gating of edges.
> Attention can also be used to explicitly select a subgraph of interest that is then used for classification [Miao2022], a little bit more akin to what happens in our model. However, in all these cases the attention is computed over all edges and the overall model is usually even less expressive than GIN (1-WL) [Xu2019] due to the (weighted) mean pooling. Because of this, GIN is usually the common baseline for graph classification. Another category is the transformer models for graphs [Kreuzer2021] which achieve impressive results, but often use dense attention between all nodes (\\(O(n^2\\)), which goes counter to our model’s goal to allow for low computational complexity and more local exploration of the graph structure. They also often depend on pre-computed features for expressive power, while in our approach we aim to do without them.
>
> Also, as shown in Table 1, using other kinds of probabilistic transitions can be a viable choice under our model. Even though a slightly lower performance is expected such a case.
>
> To reiterate, we see the main contribution of our paper as presenting a model that is both expressive (even more so than random-walk models) and offers good computational complexity.
> Because of this, we are unsure, what would even be an appropriate experimental comparison to attention-based models. Transformers of course offer better results, but their computational complexity is high and they often depend on pre-computed features for expressive power [Kreuzer2021], while models such as GAT are conceptually quite close to GIN and are not expressive. This is why we chose to compare with other expressive baselines instead.
>
> [Veličković2018] Petar Veličković, Guillem Cucurull, Arantxa Casanova, Adriana Romero, Pietro Liò and Yoshua Bengio. Graph Attention Networks. ICLR (2018)
>
> [Miao2022] Siqi Miao, Miaoyuan Liu and Pan Li. Interpretable and Generalizable Graph Learning via Stochastic Attention Mechanism. ICML (2022)
>
> [Xu2019] Keyulu Xu, Weihua Hu, Jure Leskovec and Stefanie Jegelka. How powerful are graph neural networks? ICLR (2019)
>
> [Kreuzer2021] Devin Kreuzer, Dominique Beaini, Will Hamilton, Vincent Létourneau and Prudencio Tossou. Rethinking graph transformers with spectral attention. NeurIPS (2021)

---

> ### Author Response · Authors · 2022-11-19
> **Response to reviewer wMRf (1/6)**
>
> We would like to thank you for your review and remarks. We address your questions and concerns in the subsequent comments and below.
>
> >While the idea of intelligent walk is interesting, the language of agent is not useful. The single agent case is just a single initialization of an intelligent walk with memory and multi agent case is its extension to multiple walks. It is also confusing to present polling as some sort of global communication, it is just better to present it as aggregation of information from multiple walks.
>
> We are glad you found the core idea interesting. The purpose of using the word agent was chosen precisely to signify that graph exploration is made intelligently and in parallel. Please note, that besides having memory and performing an intelligent walk, the agents also update the graph as they go. As we discuss in our theory section, this part is necessary for good expressive power (better than 2-WL) as also is the neighborhood inspection/aggregation (please see Appendix I). We feel that calling the approach just an intelligent walk would effectively discard these two parts that are actually very important.
>
> As for pooling and global communication, these are actually two different things/steps in the model. The pooling that is used for making the final prediction as you say indeed just aggregates information from all of the different agents/walks and produces a graph-level prediction. Now what we refer to as global communication and mostly discussed in Appendix C.2 is an optional extension that indeed allows for information transfer between all agents. In this case at every time step, when updating nodes (and thus also agents that are on them) we can include the mean of all the agent embeddings (not only ones that are on the node) as input to the node update MLP (after node update the node state is used to update each agent that is on the corresponding node). This way information is transferred even between faraway agents and is used to update the nodes and the agents and not for the readout/making the graph-level prediction. This mean pooling of course is a very simple global information transfer between the agents, compared to, say, full attention. But it is computationally inexpensive and as shown by [Segol2020], sum pooling on sets is expressive enough to achieve universality (since we use a fixed number of agents, mean ~ sum). We will try to make the distinction between these two different parts of the model clearer in the paper.
>
> [Segol2020] Nimrod Segol and Yaron Lipman. "On universal equivariant set networks." ICLR (2020).

---

### Author Response · Authors · 2022-11-19
**General comment**

We would like to thank all of the reviewers for their thorough reviews. We will answer all of your questions individually, but first some general remarks.

We have included more experiments and ablations in Appendices G-K, besides the previously existing extra experiments on QM9 in Appendix F. We also included the state-of-the-art random walk based GNN [Toenshoff2021] as another baseline.

[Toenshoff2021] Jan Toenshoff, Martin Ritzert, Hinrikus Wolf, Martin Grohe. "Graph Learning with 1D Convolutions on Random Walks" ArXiv 2021

A technical note regarding the Weisfeiler-Leman (WL) hierarchy: this is a central concept in the study of GNN expressiveness, also mentioned several times in our work. Unfortunately, there are two different definitions of this hierarchy in the literature, leading to slightly different indexing: the "classical" version and the "folklore" version (the latter is sometimes denoted by FWL). For any k>1, the classical (k+1)-WL is equivalent to the folklore k-WL in terms of expressiveness; for example, the Rook's 4x4 / Shrikhande graphs are the canonical example for being indistinguishable by 3-WL under the classical WL, and by 2-WL under the folklore WL. This has caused confusion in the past; see e.g. [Huang2021] for more details on the topic.

Originally, we used the classical WL indexing in our paper; however, recent works focusing on GNN expressiveness tend to use the folklore version more often (e.g. the work of Geerts mentioned by Reviewer pbyu), and some of our reviewers also used the folklore indexing in their reviews. Hence to avoid any misunderstanding, we have now switched to the folklore indexing both in the paper and in our answers for the review discussions.

[Huang2021] Ningyuan Teres Huang and Soledad Villar. "A Short Tutorial on The Weisfeiler-Lehman Test And Its Variants." ICASSP 2021

---

### Decision · Program_Chairs · 2023-01-20

**Decision:**

Accept: poster

**Justification For Why Not Higher Score:**

The AC recommends authors to continue addressing some of the newly raised points by reviewers, such as the real-world usage of the proposed method and providing more context to better understand the expressivity of the proposed method.

**Justification For Why Not Lower Score:**

The authors did a nice job in response, adding more experiments and ablations and strengthening/clarifying theoretical analysis. The overall attitudes of reviewers are positive.

**Metareview: Summary, Strengths And Weaknesses:**

This paper introduces an agent-based GNN which differs fundamentally from the message-passing based architecture. The proposed approach modifies the existing message passing mechanism by introducing the concept of an agent(s) capable of aggregating different types of graph (node, edge) information at subgraph level and then mapping it to graph level output. The key difference from typical GNNs is that agents maintain their own global state as they explore the graphs, while also being able to "read" the other agents' contributions to the node states. The author studies the theoretical analysis of proposed agent based method in counting substructures, for both 1 agent case and multi agent case. The main theoretical results revolve around the ability of the new approach to distinguish non-isomorphic subgraphs and to count various substructures such as cycles and cliques. The authors show that their model is more expressive than message-passing and even higher-order GNNs, while also having a cost sublinear in the number of nodes. Through both synthetic data and real-world data experiments, the authors show how the model is competitive with other baselines and confirm its enhanced expressive power. The authors did a nice job in response, adding more experiments and ablations and strengthening/clarifying theoretical analysis. The AC recommends authors to continue addressing some of the newly raised points by reviewers, such as the real-world usage of the proposed method and providing more context to better understand the expressivity of the proposed method.

**Note From Pc:**

if the above contains the word "oral" or "spotlight" please see: "oral" presentation means -> notable-top-5% and "spotlight" means -> notable-top-25%. As stated in our emails, we are disassociating presentation type from AC recommendations